# Characterization of Pyrolytic Tars Derived from Different Biomasses

Paula Saires , Cindy Ariza Barraza, Melisa Bertero *, Richard Pujro, Marisa Falco and Ulises Sedran

Instituto de Investigaciones en Catálisis y Petroquímica "José Miguel Parera" INCAPE (UNL–CONICET), Colectora Ruta Nac.168 Km 0, Santa Fe 3000, Argentina; psaires@fiq.unl.edu.ar (P.S.); skarlett@fiq.unl.edu.ar (C.A.B.); rpujro@fiq.unl.edu.ar (R.P.); mfalco@fiq.unl.edu.ar (M.F.); usedran@fiq.unl.edu.ar (U.S.)
* Correspondence: mbertero@fiq.unl.edu.ar; Tel.: +54-9-342-5246489

**Abstract:** The pyrolysis of three different biomasses, rice husk (RH), zoita wood sawdust (ZW) and pine wood sawdust (PW), was studied at 500 °C in a multipurpose unit at the bench scale to determine the yields of the different products and the compositions and properties of the liquid products, with particular emphasis given to the alquitranous fractions (tars). It was possible to link the characteristics of the tars with the compositions of the raw biomasses and verify their potential in various applications. The analytical techniques employed in the characterization of biomasses included lignin, cellulose and hemicellulose analysis, ultimate and proximate analysis and thermogravimetry–mass spectrometry analysis (TG-MS). Elemental analysis, gas chromatography–mass spectrometry (GC-MS), nuclear magnetic resonance spectroscopy ($^1$H NMR), Fourier transform infrared spectroscopy (FTIR) and size exclusion chromatography (SEC) were used to characterize the tars. The tar yields were 1.8, 7.4 and 4.0 %wt. in the cases of RH, ZW and PW, respectively. The tars showed higher carbon content, between 60.3 and 62.2 %wt., and lower oxygen content, between 28.8 and 31.6 %wt., than the corresponding raw biomasses. The main components of the tars had aromatic bases, with phenols representing more than 50%. Tar RH included more guaiacols, while Tars ZW and PW included more phenols and alkylated phenols.

**Keywords:** pyrolysis; tar; phenols; lignocellulosic biomass



## 1. Introduction

The use of biomass resources to produce chemicals and energy is being encouraged due to limitations concerning both availability and environmental issues in the global energy system, which is based on fossil resources. Thus, the search for alternative, renewable sources, has been reinforced in public agendas everywhere, aiming for a reduction in the dependency on fossil fuels and in the associated negative social, environmental and economical consequences derived from their utilization. Biomass has an important share in the world energy matrix, about 10%, or even higher in countries such as Sweden, Finland and Denmark; however, this contribution differs markedly among the various world regions [1].

Using lignocellulosic biomass to obtain energy or chemicals has various advantages, among which are (i) local availability, (ii) the possibility of adding value to residues of the agricultural and forestry industries, (iii) higher production efficiency, (iv) the conversion of environmental liabilities (residues, effluents) into raw materials and (v) the redistribution of income to rural activities. Starting with lignocellulosic biomass, solid, liquid and gaseous fuels, which can be used in various applications, can be obtained by means of processes with different levels of sophistication. However, as the energy density of this material is low and it is widely dispersed, supply logistics represents a significant portion of the total production costs (from 33 to 50%) [2]. Then, decentralized, mobile plants are preferentially designed after obtaining knowledge of biomass availability and distribution.

Under this approach, residues from agricultural crops and primary industrial activities, such as grain and bean grinding, as well as forestry residues, are potentially exploitable [3]. Depending on the country or region, some cases are particularly important in magnitude and show a small number of applications, for example, rice husks (representing about 22% of rice yield), with a global annual yield of more than 600 millions tons [4]. Presently, rice husks are burnt to produce energy, used in chicken farming or added to concrete (pozzolanic addition) [5]. However, given their high silicon content, the uncontrolled combustion of rice husks generates health and environmental problems as particles remain suspended in the air [6]. Rice husks could be used as a raw material in thermochemical processes, producing energy and chemicals, thus reducing the disposal problem [7]. Moreover, as this material is generated only in rice mills, storage, transportation and conditioning costs are reduced substantially.

Analogously, large amounts of residues from the forestry industry, such as wood sawdusts, are produced close to forestry exploitations, as well as in processing mills. Studies exist that point to the potential use of forestry residues to produce energy, and the best results in terms of energy yields and lower impacts of greenhouse gases are obtained when the residues are raw materials, as specific crops are not needed [8]. Great potential exists in many countries to develop bioenergy projects based on forestry biomass, given the strong economic relevance of their forestry industries. However, it is necessary to analyze the management options of forestry resources soundly, in order to achieve their sustainable utilization.

Pyrolysis and gasification processes could use these residual biomasses. Three useful bio-products are produced in pyrolysis, where biomass is thermally degraded in the absence of or under limited amounts of oxygen: liquids (named bio-oils, composed of an aqueous phase and a viscous phase, usually named tar), gases and solids (char, basically formed of carbon and inorganic material). The yield and composition of the different streams depend on the operative process parameters and the raw biomass. Fast pyrolysis maximizes the yields of gases or liquids [9]; for example, at 425–500 °C, the liquid yield can be as high as 75 %wt., while at 700 °C, gases are maximized. Conventional pyrolysis, running at 500–600 °C, with more extended contact times, yields gases, liquids and solids with similar proportions.

A techno-economic analysis showed that the production of transportation fuels from biomass pyrolysis had economic advantages over the gasification and biochemical conversion pathways [10].

In gasification, which is conducted at temperatures higher than 800 °C, including an oxidizing agent (usually air) at concentrations lower than stoichiometric ones, tars present in the producer gas represent a major technological problem, as for certain applications, such as internal combustion engines, they must be removed. This can be achieved by a thermal effect only or by means of metallic or zeolite catalysts [11].

Jerzak et al. [12] studied the pyrolysis of different types of agro-industrial waste (medium-density fiberboard, brewery bagasse and post-process soy flour from oil extraction) to assess the energy requirements. It was found that 4–10% of the higher heating value of these raw materials was missing, preventing them from achieving the self-sustaining energy of intermediate pyrolysis. Bio-oils and their constituting fractions find multiple applications and valorization opportunities. For example, the aqueous fractions could be converted to hydrocarbons by means of hydrotreating [13] and/or cracking over acidic catalysts [14], be the source of valuable chemicals [15] or be used as wood-conserving agents [16].

Tar can be considered more versatile than the aqueous fraction, as it contains much less water and is less acidic. However, the possibilities for the valorization of tar have not been explored exhaustively, as the complexity of the mixture makes its characterization difficult [17]. Tars from pyrolysis are mainly composed of monophenols such as phenol, guaiacol, syringol and their derivatives, and oligomeric phenols such as stilbene, phenyl-coumarin and resorcinol [17]. Moreover, in cases where the raw biomass includes proteins

(in cases of cereal shells or cow manure), the pyrolytic derivatives of those compounds, which include nitrogen, preferentially concentrate in tars [18].

Some possible uses for pyrolysis tars, which can be easily separated from the aqueous fractions by decanting, are the partial substitution of phenol in the production of phenol-formaldehyde resins [19] and of polymers in the obtention of spheres supporting slow-release agrochemicals and biocides, taking advantage of their biodegradability [20]. The addition of tars to asphalt binders, partially replacing more valuable fossil resources and also playing the role of antioxidants preserving asphalts, is similarly very attractive [21]. The co-processing of bio-oils in catalytic cracking (FCC) units, that is, taking parts of mixtures with standard hydrocarbon feedstocks, has been studied in recent years [22], with the catalysts and process conditions being adequate. However, a number of issues remain to be solved, such as the impact of water from bio-oils [23] and the occurrence of high concentrations of oxygenated compounds.

All these possible uses of tars require the exploration of process conditions maximizing their yields, as well as an understanding of the mechanisms by which their many components are formed and the variations in tar composition and their properties as a function of raw biomass. Moreover, it is absolutely necessary to characterize tars completely to design more efficient valorization processes. Under this view, the properties of three different biomasses (rice husk and two wood sawdusts), as well as their performance in pyrolysis, were determined, following their decomposition and the formation of organic compounds as a function of temperature. The yields of each product stream were obtained and the liquid and gas fractions were characterized, with emphasis placed on tar, using various techniques. The differences in tar composition and properties were highlighted and linked to the composition of the raw biomasses.

The objective of this work is to carry out careful characterization of tar derived from different agro-industrial residual biomass, taking into account the possible applications of these liquid products.

## 2. Materials and Methods

### 2.1. Biomass Characterization

The rice husk (RH) biomass was provided by a rice mill in the Santa Fe province, Argentina, with the particle size being in the 1.7–2.4 mm range. The wood sawdust from the forestry industry were derived from pine (*Pinus elliotti*, PW) and zoita (*Luehea divaricata*, ZW) woods, provided by sawmills in the area close to Santa Fe city, Argentina, with a particle size in the range of 1.7–4.7 mm. The contents of water, volatile matter and ash were determined according to the ASTM D 3173 [24], D 3175 [25] and ASTM D 3174 [26] standards, respectively. The amount of fixed carbon was determined by calculating the difference. The elemental compositions of the raw biomasses were determined with CHN628 Series Elemental Determinator (LECO) equipment. The higher heating values (HHVs) were determined following Dulong's formula [27], based on the elemental compositions.

$$\text{HHV}(\text{MJ kg}^{-1}) = 0.3383 \cdot \text{C} + 1.443\left(\text{H} - \frac{\text{O}}{8}\right)$$

where, C, H and O are the mass percentages of carbon, hydrogen and oxygen, respectively, on a dry bases.

The compositions of the raw biomasses were determined according to the parameters Acid Detergent Lignin (ADL) (PROMEFA V2 protocol in ANKOM equipment (ANKOM Technology, Macedon, NY USA)), Sequential Acid Detergent Fiber (ADFS, ISO 13906:2008) [28] and amylase Neutral Detergent Fiber (aNDF, ISO 16472:2006) [29], considering that ADL represents the amount of lignin in the biomass, ADL S the addition of lignin and cellulose, and aNDF the addition of lignin, cellulose and hemicellulose [30].

Thermogravimetry–mass spectrometry analysis (TG-MS) of the raw biomasses was performed using a thermogravimetric analyzer (TGA 2950 TA instrument, TA Instruments Inc., New Castle, DE, USA) and a mass spectrometer ThermoStar Pfeiffer (Pfeiffer, Milpitas,

CA, USA). The furnace temperature for TGA was increased from room temperature to 1000 °C under a nitrogen flow of 100 mL min$^{-1}$ at a heating rate of 10 °C min$^{-1}$, and the mass-to-charge ratio was determined using a multi-channel ion detection system.

## 2.2. Pyrolysis Experiments

The experiments of pyrolysis were performed in a multipurpose unit at the bench scale. The unit, which is schematized in Figure 1, is a reactor which can be operated in the gasification or pyrolysis mode depending on minor operative adjustments. The downwards flow induces the circulation of tar through the hottest fraction of the reactor, thus decreasing its concentration during the operation in the gasification mode. Moreover, air easily contacts biomass, without the usual problems that occur in throat reactors. The reactor length is 120 cm, the diameter is 20 cm and the useful volume is 25 L. Biomass was fed from a 25 L hopper by means of a variable pitch screw conveyor, with the flows being from 1 to 1.2 kg h$^{-1}$. Solid products (char, basically formed of carbon and inorganic material) were continuously removed from the reactor through a screen plate by means of a rotating paddle and weighed to assess their yield. Ignition in the reactor was produced by the combustion of propane and then it was operated autothermally. The reaction temperature in the experiments was 500 °C (as controlled in position T1) and the flow of air was 16 L min$^{-1}$. Liquid products were collected as follows: the aqueous phase was recovered from a closed vessel at the cyclone exit with natural condensation (point 6 in Figure 1) and the tar phase at the condenser exit (point 7 in Figure 1); both phases were weighed for quantification. The gas phase was filtered through a glass fiber filter and its flow was recorded. The duration of each experiment was 5 h, and the mass balances (recoveries) were over 95%.

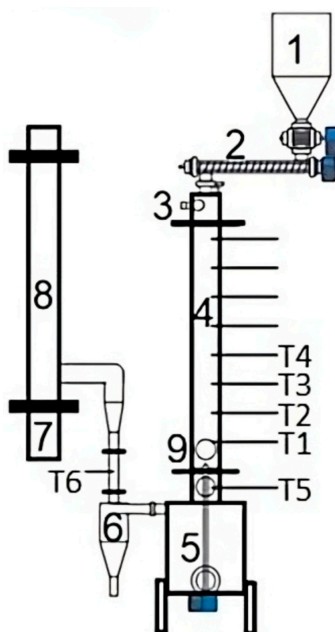

**Figure 1.** Pilot-scale pyrolyzer/gasifier. (1) Biomass hopper; (2) screw conveyor; (3) air inlet; (4) reactor; (5) solid collector; (6) cyclone; (7) tar collector; (8) condenser and gas exit; (9) ignition port; T1 to T6, thermocouples.

## 2.3. Product Analysis

The aqueous phase was analyzed by conventional capillary gas chromatography using Agilent 6890N equipment (Agilent Technologies, Santa Clara, CA, USA)with flame ionization detection (FID) using an HP-5 column measuring 30 m long with a 0.32 mm internal diameter and a 0.25 μm phase thickness. Product identification in the GC analysis was performed by mass spectrometry using Shimadzu GCMS-QP2020 equipment (Shimadzu Latin America S.A., Montevideo, Uruguay). The calibration of chromatographic areas was

performed with response factors for each of the chemical groups. Mass response factors were determined by using mixtures of standard compounds which are representative of different chemical groups in pyrolysis liquids. Tetraline was used as a reference compound, with its response factor being one. The selected standard compound for acids and esters was acetic acid; for aldehydes, acetaldehyde; for ketones, acetone; for alcohols, methanol; for phenols, phenol; for ethers, 1,2,4-trimethoxybenzene; and for nitrogen-containing compounds, pyridine. Unidentified peaks, each representing less than 0.5% of the total chromatographic area, were assigned an average response factor that was calculated with those of the compounds identified in the same range of elution times.

The relative mass response factors were calculated with the following equation:

$$\frac{m_i}{m_{st}} = f_i \frac{A_i}{A_{st}}$$

where $m_i$ is the mass of the compound i, $m_{st}$ is the mass of the reference compound, $f_i$ is the relative chromatographic response factor of the compound i and reference compound, $A_i$ is the chromatographic area of the compound i, and $A_{st}$ is the chromatographic area of the reference compound.

The viscous phase (tar) was analyzed by gas chromatograph with mass spectrometry, using a Shimadzu GC-2030 high-performance gas chromatograph with an SH-Rxi-5Sil MS column (5% polar) measuring 30 m long with a 0.25 mm internal diameter and the 0.25 μm phase thickness, and a Shimadzu GCMS-QP2020NX mass detector. Analyses were performed in split mode using a helium gas carrier at a 1.0 mL min$^{-1}$ flow. The mass range for data collection was 40–400 Da with a 0.3 s scan interval. Tar was dissolved at 1% into methylene chloride for injection in GCMS.

The content of water in the aqueous phase and tar was assessed by means of Karl Fischer titration (IRAM 21320) [31], and the density and pH of the aqueous phase were determined by a conventional volume–mass method and with a HANNA HI 8424 pH-meter (Hanna Instruments, CABA, Argentina), respectively. The elemental composition of tar was determined using CHN628 Series Elemental Determinator (LECO) equipment (LECO Corporation, St. Joseph, MI, USA). The higher heating values (HHVs) of tars were determined with Dulong's formula [27].

Tar samples were also subjected to Fourier transform infrared spectroscopy (FTIR), nuclear magnetic resonance spectroscopy ($^1$H NMR) and size exclusion chromatography (SEC). FTIR analyses were performed in a Shimadzu FTIR Prestige-21 spectrometer with a high-sensitivity detector (400–4000 cm$^{-1}$, 80 scans) (Shimadzu Latin America S.A., Montevideo, Uruguay), with the samples prepared as wafers. The samples were diluted with KBr at a 1:100 mass ratio and mixed in a mortar to form wafers about 13 mm in diameter and 1 mm in thickness. $^1$H NMR spectra were recorded in a Multinuclear Bruker Avance 300 MHz digital spectrometer (Bruker Corporation, Cordoba, Argentina) at 25 °C in CDCl$_3$.

The size exclusion chromatography SEC analyses were performed in a Waters chromatograph with a Waters 2414 differential refractometer detector, using tetrahydrofuran as the mobile phase with a 1 mL min$^{-1}$ flow, a 50 μL injection volume and a 15 min running time. The calibration was performed with mixtures of polyethylene glycol standard and polystyrene, which allowed us to determine the characteristics of the molar mass (MM), that is, the average molecular weight (Mw, calculated from the weight fraction distribution of differently sized molecules) and number average molecular weight (Mn, calculated from the mole fraction distribution of differently sized molecules in a sample and peak molar mass (Mp), and dispersity (D = Mw/Mn)).

## 3. Results

### 3.1. Composition of Raw Biomasses

The compositions and some properties of the raw biomasses are shown in Table 1, where it can be seen that their content of water is appropriate to be used in thermochemical processes. The elemental compositions and the amounts of lignocellulosic material are

typical of wood sawdusts [32,33] and grain shells [34]. However, important differences between the various biomasses can be noticed: the contents of cellulose and lignin in rice husk are smaller than in wood sawdusts, and, on the contrary, the content of ash in rice husk is high and, consequently, its volatile matter is low, associated with less C and O. The high content of ash in rice husk is typical, as extensively reported in the literature (for example, 17 %wt. [35], 16.5 %wt. [36], 19.3 %wt. [37]), with silica being the main component with about 87% [38] to 95% [39]. Nitrogen is found to be present in small amounts, less than 1 %wt. in all cases; this is adequate in relation to the production of pyrolysis syngas and oils, as this element increases pollution after their use [40]. HHVs are also commonly found in these biomasses; for example, from 16.7 to 21.3 MJ kg$^{-1}$, HHVs were reported for pine sawdust [41,42] and 16.6 MJ kg$^{-1}$ for rice husk [36].

**Table 1.** Biomass composition and properties.

|  | RH | ZW | PW |
|---|---|---|---|
| Moisture (%wt.) | 8.6 | 6.8 | 6.6 |
| Proximate analysis (%wt., dry basis) | | | |
| Ash | 23.2 | 2.9 | 1.1 |
| Volatile matter | 60.9 | 79.9 | 78.9 |
| Fixed carbon | 15.9 | 17.2 | 20.0 |
| Ultimate analysis (%wt., dry basis) | | | |
| C | 39.2 | 52.5 | 54.0 |
| H | 4.4 | 6.3 | 6.3 |
| O [a] | 32.9 | 37.9 | 38.2 |
| N | 0.3 | 0.4 | 0.4 |
| Component analysis (%wt., dry basis) | | | |
| Cellulose | 37.8 | 56.1 | 56.9 |
| Hemicellulose | 18.0 | 12.6 | 12.2 |
| Lignin | 17.3 | 25.0 | 28.4 |
| Other [b] | 3.7 | 3.4 | 1.3 |
| Higher heating value (HHV, MJ kg$^{-1}$) | 13.7 | 20.0 | 20.4 |

[a] 100-C-H-N-Ash; [b] 100-cellulose-hemicellulose-lignin-ash.

The distributions of lignocellulose components in sawdusts were similar and consistent with previously reported values [43]. Rice husk composition was in line with other reports: 25–40% cellulose, 8–21% hemicellulose, 15–31% lignin, 15–17% ash and waxes, 2–8% other soluble substances and up to 3% proteins [44,45]. Particularly, Costa et al. [46] studied the contents of the lignocellulosic components of zoita sawdust to evaluate its potential as a pyrolysis raw material and reported 23.5 %wt. lignin, 68.1 %wt. holocellulose, 6.4 %wt. extractives and 1.9 %wt. ash. In the case of pine sawdust, which is well known, Rusanen et al. [47] reported 44 %wt. cellulose, 26 %wt. hemicellulose, 26 %wt. lignin, 3 %wt. extractives and 1 %wt. ash.

### 3.1.1. Polymeric Structure of Biomass Components

Lignocellulosic biomasses are composed of polysaccharides, phenolic compounds and other minor constituents, such as minerals, lipids, extractives, etc. Polysaccharides are cellulose, hemicellulose and pectins, while phenolic compounds are part of the lignin structure.

Cellulose is a crystalline homopolymer also including some amorphous zones, with its structure identical in all the materials which contain it. It is formed of lineal chains of cellobiose (D-glucopyranosyl-β-1,4-D-glucopyranose), having between 10,000 and 15,000 glycosidic units [48], that is, only glucose units linked through ether bonds [49].

On the contrary, hemicellulose is formed of amorphous heteropolysaccharides, with between 100 and 200 units, with the units and the structure differing between biomasses according to their source. For example, in the cases of hard woods and many agricultural

products, including rice husk, hemicellulose is composed mainly of heteroxylanes (arabinoxylan or glucuronoxylan with different substitution patterns), while in soft woods, such as pine and zoita, mannans (glucomannan and galactoglucomannan) predominate [50]. Ferulic and p-coumaric acids are also present in hemicellulose [51], as well as complex phenolic compounds [52], generating ester links between polysaccharides and lignin.

Rivas et al. [53] characterized rice husk and pine sawdust hemicelluloses, observing that the former was composed mainly of xylose units (about 48 %wt., with about 93% forming oligomeric and polymeric saccharides), and glucose (about 8.3 %wt.) and arabinose (about 6.4 %wt.). Hemicellulose in pine sawdust contained mainly mannose (48 %wt.), glucose (14 %wt.), xylose (12 %wt.) and galactose (12 %wt.). The major functional groups in both hemicelluloses were uronyl and acetyl. The phenolic compounds present in rice husk hemicellulose were vanillin and p-coumaric, vanillic and ferulic acids, with syringic acid also observed in pine sawdust.

Pectins are polysaccharides, mostly constituting galacturonic acid, and behave as adhesives for hemicellulose [49]. Pectins in wood sawdusts also include rhamnose and arabinose [54].

Lignin is a biopolymer which is mainly composed of three repetitive units, coumaryl (H), guaiacyl (G) and syringyl (S) units, which derive from their respective monolignols, that is, p-coumaryl (4-hydroxycinnamyl), coniferyl (4-hydroxy-3-methoxycinnamyl) and sinapyl (4-hydroxi-3,5-dimethoxycinnamyl) alcohols. These units are linked by ester, ether and carbon–carbon bonds; moreover, lignins which include fewer C–C bonds are less condensed and, consequently, can be degraded more easily. S units have two aromatic substitutions with a methoxy group, while G units have only one methoxy group, thus forming more C–C bonds [48].

Lignins from herbaceous species such as rice include more H units than woods and show a higher degree of acylation with acetate groups, p-coumaric acid and/or p-hydroxybenzoate species [55], with their resulting structure being more linear than that in woods. Lignin in rice husk is mainly formed of G units, with an H/G/S ratio of 7:81:12, mostly including β-O-4′ alkyl-aryl ether units (representing 65% of all inter-unit linkages), but also including other condensed units such as phenylcoumarans (23%), dibenzodioxocins (5%), resinols (4%) and spirodienones (3%), as well as cinnamyl alcohol (6%) and cinnamaldehyde (5%) end-groups. Moreover, it is partially acylated (approximately 10–12%) in the γ-OH group of the side chain in p-coumarates, preferentially in the S units [56].

The three biopolymers constituting lignocellulosic biomass intercross by means of different types of bonds forming complexes, in such a mode that they can be separately considered cellulose, hemicellulose and lignin fractions. Dehydroxyferulate bridges are found more frequently at polysaccharide intercrossings, while ferulates and p-coumarates can couple polysaccharides with lignin, being predominant in herbaceous biomasses such as rice [57]; woods, on the contrary, show mainly benzyl ether, benzyl ester, phenyl glycoside and acetal links [58].

The minor components in lignocellulosic biomass are extractable compounds, proteins and ash. Extractable compounds are hydrophilic (free phenolic compounds such as stilbene, oligolignols and derivatives, tannins and flavonoids) or lipophilic (free fatty acids and alcohols, resin acids, hydrocarbons, terpenoids, steroids and sitosterols). Rosado et al. [56] reported extractives in acetone (4.3 %wt.), methanol (1.9 %wt.) and water (4.5 %wt.) in rice husk, with tricin being the most important flavone. Tanaka et al. [59] characterized extractions using various solvents (methanol, chloroform, ethyl acetate) in zoita wood and observed triterpenes (such as maslinic acid), flavones (such as vitexin, a C-glycoside flavone), steroids (such as glucopyranosylsitosterol) and flavonoids (such as epicatechin). Ren et al. [60] characterized pine wood and reported acetone extractives in amounts from 1.5 to 4.8 %wt. and methanol extractives up to 11.2 %wt. It can be observed in Table 1 that the content of minor components in rice husk, particularly ash and others, was much higher than in wood sawdusts.

### 3.2. Thermogravimetric Analysis

The results of the thermogravimetric analyses of the various biomasses are shown in Figure 2, where it can be seen that the profiles are typical of forestry [61] and agricultural [62] lignocellulosic materials. The thermal decomposition of biomass involved three steps in all the cases. In the first step, from room temperature up to about 100 °C, the loss of water and some volatiles represented about 2 %wt. for the three materials. The most important mass loss was observed for all the cases during the second step, occurring from 200 to 400 °C, where 46% of the rice husk and more than 70% of the wood sawdusts decomposed. Two thermal events manifested during this step: the first starting at 200 °C with a maximum temperature of 300 °C, and the second one from 325 °C with a maximum between 345 and 360 °C, corresponding to the decomposition of hemicellulose and cellulose, respectively [63]. It was observed in the case of rice husk that the rate at which the first event develops is lower than that of the second event, thus producing a wider peak, as was also observed by other authors in purely thermal processes [64] and catalytic pyrolysis [65]. Most of the lignin and part of the residual carbonaceous material decomposed slightly during the third step, initiating at about 400 °C. Finally, the undecomposed residual material corresponds to ash and fixed carbon (see values in Table 1). These results are consistent with reports by other authors for pine sawdust [66] and rice husk [35].

The thermal degradation studies performed on the main components of lignocellulosic materials showed that hemicellulose decomposes between 220 and 315 °C, and cellulose between 315 and 400 °C [67], with lignin covering a much wider temperature range, from approximately 280 to 1000 °C [68].

Particularly, the thermogravimetry of rice husk has been extensively studied; however, a few reports show the products evolving as a function of time and temperature during pyrolysis. Thermogravimetry–mass spectrometry (TG-MS) shows the real-time mass spectra of species generated during pyrolysis, thus allowing the observation of species formed as a function of time. Figure 3 shows the time evolution of $m/z$ ratios as a function of temperature for the three biomasses, that is, water ($m/z$ 18), methane ($m/z$ 15), carbon dioxide ($m/z$ 44), furfural ($m/z$ 96), acetic acid ($m/z$ 45) and methanol ($m/z$ 31). It is not possible to follow the evolution of carbon monoxide as its characteristic $m/z$ ratio of 28 is coincident with that of nitrogen, the carrier gas.

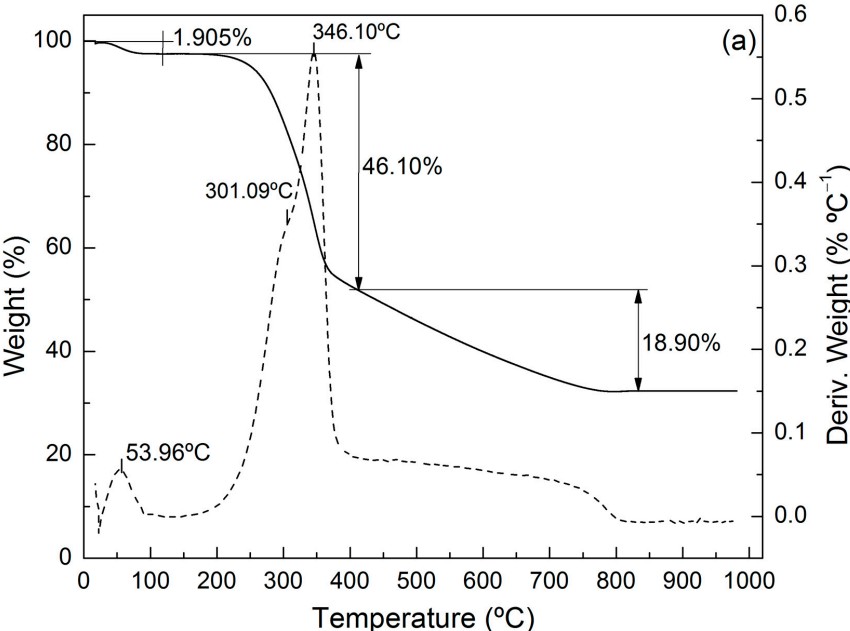

**Figure 2.** *Cont*.

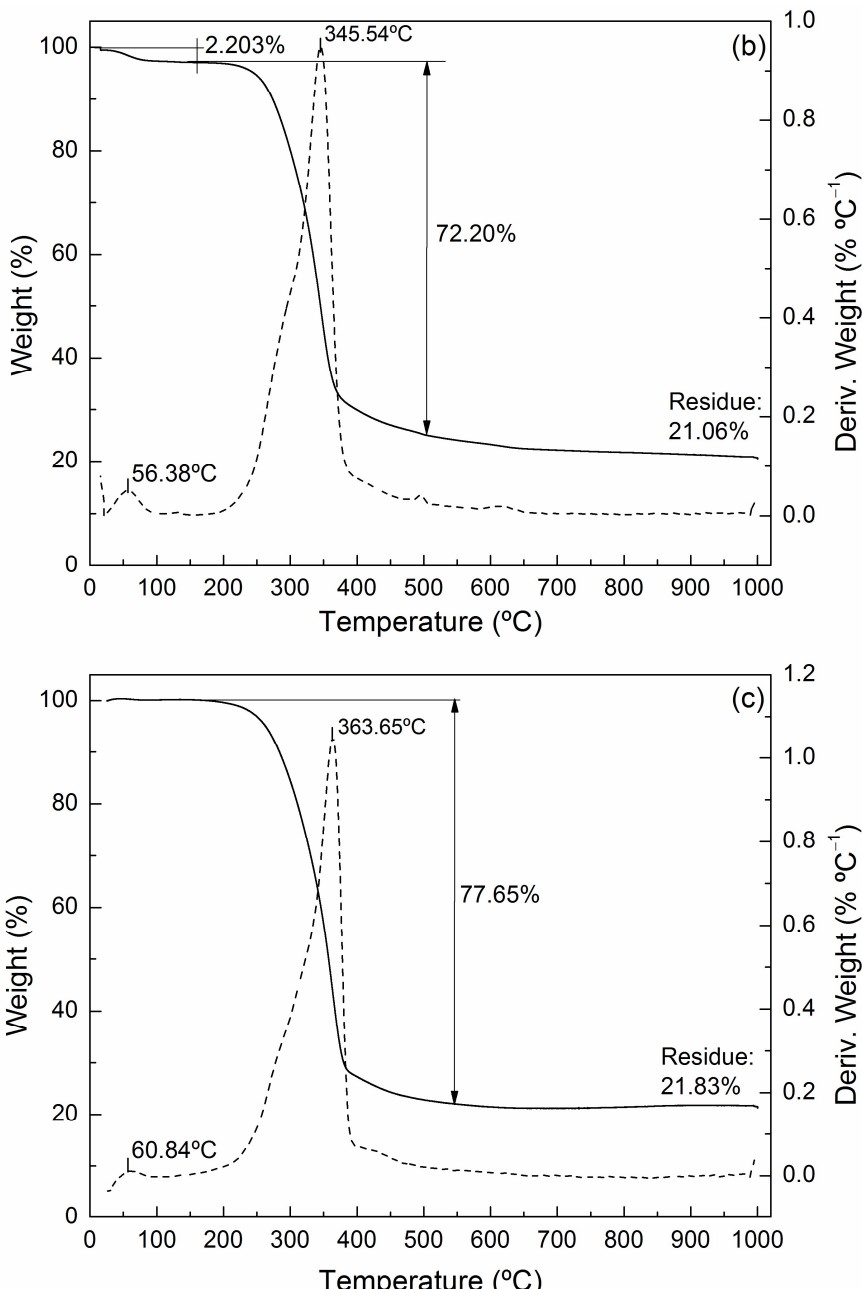

**Figure 2.** TG and derived TG curves. (**a**) Rice husk; (**b**) zoita wood sawdust; (**c**) pine wood sawdust.

It can be observed for the three biomasses that water is produced during the whole pyrolysis temperature range, notably decreasing at temperatures over 600 °C, consistent with observations by Muley et al. [69], who studied the pyrolysis of pine sawdust and its fractions cellulose and lignin. Two peaks were observed in the yield of water, as shown in Figure 2: the first corresponds to the loss of humidity in the materials at about 100 °C, and the second shows a maximum at about 360 °C, corresponding to the most important mass loss shown in Figure 2, which can be attributed to dehydration reactions of various components in biomass. During a thermogravimetric analysis of hemicellulose and lignin, Luo et al. [33] observed that the number of C–OH groups, as quantified by the percentage area in FTIR spectra, decreased from 36 to 24.3% in the case of hemicellulose and from 65 to 50% in the case of lignin when the temperature increased from 110 to 230 °C, a fact which was attributed to dehydration reactions. In the case of rice husk, the first peak extended up to 150 °C, suggesting the release of water through a method other than evaporation, such as, the scission of glycosidic bonds in cellulose, which are favored at temperatures lower than

200 °C with low heating rates [49]. Other authors who studied the pyrolysis of residual biomasses by means of TG-MS also observed two peaks in the release of water; for example, Ischia et al. [70], in the case of sewage sludge, and Yao et al. [64], in the case of rice husk. The latter authors assigned the release of water in the 100 to 500 °C temperature range to bulk water, suggesting that at temperatures higher than 500 °C, water is formed due to cracking or reactions of the functional groups, decreasing slowly at higher temperatures. Luo et al. [33] separately studied the thermal gravimetry of cellulose, hemicellulose and lignin, observing that in the 110–230 °C temperature range, lignin lost 17.7% of its initial mass, while cellulose and hemicellulose lost less than 1.5%.

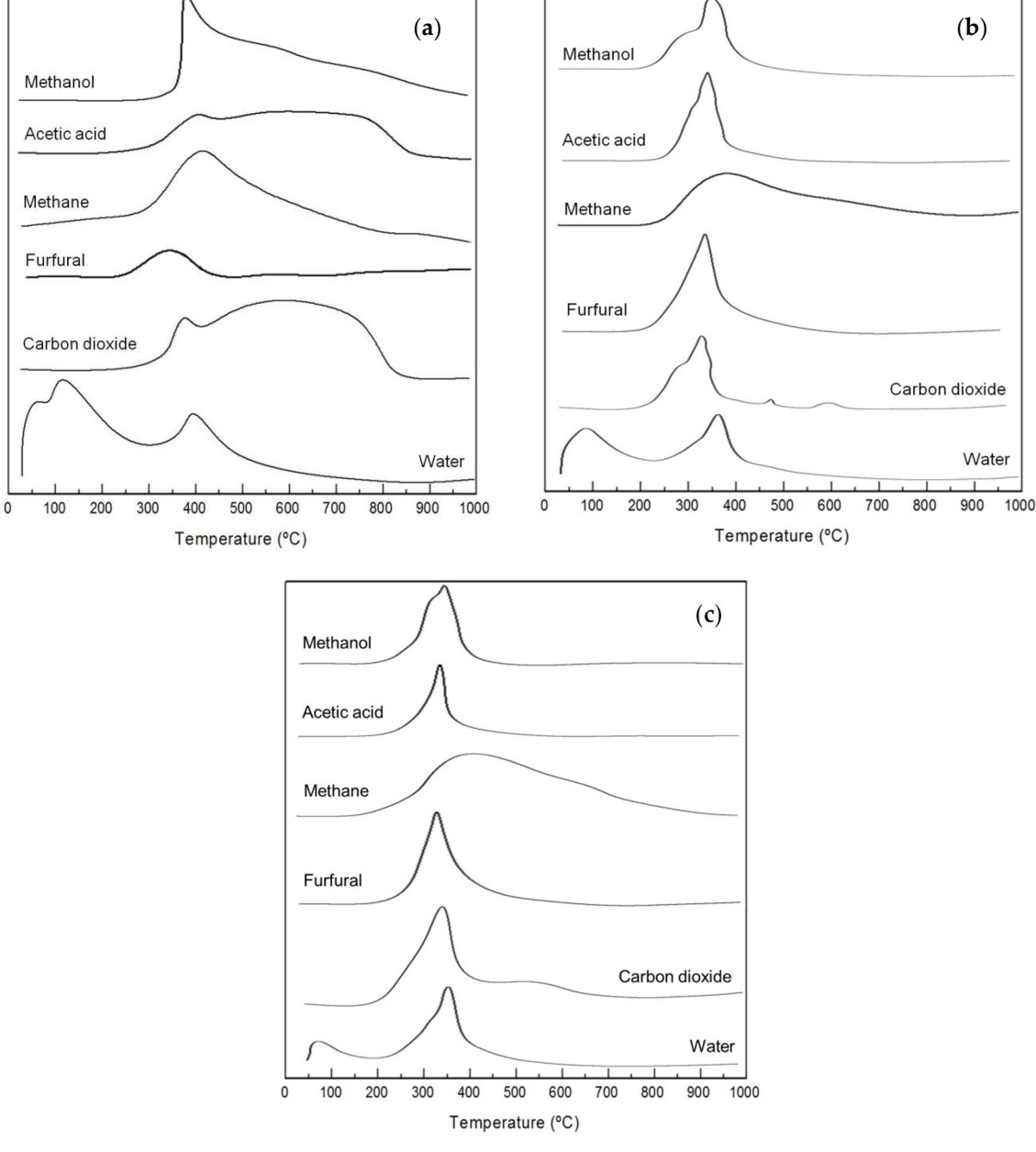

**Figure 3.** Time evolution of specific $m/z$ ratios as a function of temperature in the TG-MS experiments. (**a**) Rice husk; (**b**) zoita wood sawdust; (**c**) pine wood sawdust.

In relation to carbon dioxide, it was observed that wood sawdusts showed similar spectra, where two ranges of formation were noticed: the first one, more clearly defined and intense, between 200 and 400 °C, and the second one between 400 and 650 °C. However, the evolution of carbon dioxide from rice husk was very intense between 350 and

800 °C, with a well-defined shoulder at about 370 °C, which could be a consequence of C=O groups cracking in hemicellulose (see the composition of rice husk hemicellulose in Section 3.1.1) [67]. A similar profile for $CO_2$ generation from rice husk during TG-MS was observed by Yao et al. [64]. DeGroot et al. [54], who studied the pyrolysis of cottonwood, observed that in the first step, from 100 to 250 °C, carbon dioxide is produced by the decarboxylation of uronic acids included in hemicelluloses and pectins, with the latter decomposing completely in that temperature range. Scott et al. [71] reported the increasing production of carbon dioxide during cellulose pyrolysis in the 450 to 900 °C temperature range. Patwardhan et al. [72] observed two intervals where carbon dioxide was formed during the pyrolysis of hemicellulose isolated from switchgrass, between 250 and 400 °C, corresponding to the decomposition of carboxylic and carbonylic groups, and between 500 and 600 °C, attributable to C–O and C–C bond scissions. Yang et al. [67] noted that the pyrolysis of xylan (a major component in rice husk hemicellulose) produced carbon dioxide in the 200 to 800 °C range and, according to these authors, it can be assumed in the pyrolysis of lignocellulosic biomasses that the most important contribution to the production of carbon dioxide at temperatures lower than 500 °C comes from hemicellulose, while at higher temperatures, it is due to lignin, with cellulose contributing only at low temperatures in minor proportions.

Furfural is produced after the scission of the xylopyranose rings in hemicellulose, followed by the formation of a furanose ring and its further dehydration [72]. All the three biomasses showed a furfural peak in the 200 to 450 °C range, peaking at about 350 °C, which is consistent with results from other authors: Patwardhan et al. [72] showed that in the pyrolysis of hemicellulose from switchgrass, the formation of furfural occurs between 300 and 400 °C; Conde et al. [73] and Garrote et al. [50] demonstrated that the main products in the degradation of hemicellulose are furfural and 5-hydroxymethylfurfural from the dehydration of pentoses and hexoses, respectively.

Acetic acid is most likely formed from the removal of O-acetyl groups in mannose units included in galactoglucomannan's structure [66]. Wood sawdusts showed acetic acid peaks which were more defined than those of rice husk, with maxima located close to 350 °C, corresponding to the rapid mass loss observed in Figure 2, which can be attributed, in part, to the higher amount of acetyl groups in wood sawdust hemicelluloses (see Section 3.1.1). On the contrary, rice husk showed the continuous production of acetic acid during the pyrolysis process, starting at about 300 °C, which abruptly decreased at about 800 °C. An important contribution to acetic acid could be expected from the hydrolysis of the acetyl ester groups of hemicellulose [54].

Methanol was produced in the range from 200 to 450 °C in the cases of wood sawdusts, with a maximum about 350 °C and a shoulder close to 300 °C. Rice husk showed the continuous production of methanol starting at 350 °C, showing a maximum at about 380 °C. Methanol is formed from the rupture of methoxy groups in lignin, occurring more intensively when the temperature is lower than 600 °C [74]; at higher temperatures, the most important contribution is that from the conversion of residual char. Methanol can also be formed from the fragmentation of 4-O-methylglucuronic acid units in hemicellulose and, to a lower extent, of methyl ester groups in pectins, at temperatures lower than 250 °C [54].

The yield of methane showed wide peaks, with the maximum at about 450 °C, for all the raw biomasses. Methane is formed from the pyrolysis of phenoxymethyl units in lignin at temperatures lower than 600 °C [67]. The production of methane in wood sawdusts is more important than in rice husk in the whole range of temperatures, as sawdusts have much higher lignin content (see Table 1). According to a report by Yang et al. [67], hemicellulose, cellulose and lignin contribute to the formation of methane during biomass pyrolysis at low, medium and high temperatures, respectively. Therefore, the higher content of holocellulose in sawdusts (see Table 1) produced the highest contribution to the formation of methane at low temperatures. Scott et al. [71] observed the formation of methane in a wide temperature range from 500 to 900 °C during cellulose and maple wood pyrolysis. Above 700 °C and up to 1000 °C, methane can be formed by the gasification of

tar components, as suggested by Dufour et al. [75] in the gasification of spruce wood, or from the demethylation of aromatic hydrocarbons in tar (benzene, toluene, xylenes, phenol, indene, cresols, naphthalene, methylnaphthalenes, acenaphthylene and phenanthrene).

### 3.3. Product Yields in Pyrolysis

The most important parameters impacting biomass pyrolysis product yields are temperature, heating rate, particle size and the properties of the raw biomass. It has been extensively reported in the literature that larger particles produce more char, as poor heat transfer to the inner parts induces a lower average particle temperature and, hence, lower volatile yields [76].

Table 2 shows the product yields for the three biomasses tested. It can be seen that the wood sawdusts PW and ZW yielded more bio-oil and gases than rice husk, with the differences much more pronounced in the case of tar, a fact which can be attributed to the higher amount of cellulose in these biomasses [77]. Rice husk produced much more char than wood biomasses, consistent with its high content of ash. The higher content of hemicellulose in rice husk (see Table 1) could also contribute to the higher char yields observed with this raw biomass. Consistent with this, Aho et al. [66] showed that hemicellulose produces more char than cellulose during pyrolysis. Other authors, for example, Ben et al. [78], reported between 26 and 29 %wt. of bio-oil and between 31 and 39 %wt. of char in the pyrolysis of pine wood at temperatures between 400 and 600 °C.

**Table 2.** Pyrolysis yields (%wt.).

|  | **RH** | **ZW** | **PW** |
|---|---|---|---|
| Bio-oil | 23.2 | 28.9 | 27.1 |
|    Aqueous phase | 21.4 | 21.5 | 23.1 |
|       Water | 19.8 | 18.3 | 20.8 |
|       Organic compounds | 1.6 | 3.2 | 2.3 |
|    Tar phase | 1.8 | 7.4 | 4.0 |
| Char | 43.9 | 26.0 | 22.7 |
| Gases | 32.9 | 45.1 | 50.2 |

Bio-oils can be considered a monophasic chemical mixture; however, phase separation occurs when the content of water is higher than approximately 45% [17,79] or chemicals are present which are not miscible in the pyrolytic liquid [80]. Tar represented a higher fraction of bio-oils in the cases of wood sawdusts (15 and 26% for pine and zoita wood, respectively) than in rice husk (8%), consistent with reports from other authors for wood sawdusts, such as between 13 and 27% for lauan wood [17].

Costa et al. [46] reported char yields of 34.4 %wt. in the slow pyrolysis of zoita sawdust at 450 °C, composed of 77.2 %wt. fixed carbon, 5.2 %wt. ash and 18.1 %wt. volatile matter, with the heating value being 29.85 MJ $kg^{-1}$.

The results shown in Table 2 indicate that differences in the composition of the lignocellulosic material between biomasses strongly influenced the distribution of products from their pyrolysis, as also concluded by Biswas et al. [81] in the cases of corn cob, wheat straw, rice straw and rice husk.

### 3.4. Tar Properties and Compositions

Tar is the fraction in bio-oils containing compounds which are slightly soluble, or insoluble, in water. If water is present in bio-oil at more than approximately 30 %wt., it can be separated by gravity, a property which was used in the pyrolizer.

Table 3 shows the properties and composition of the different tars. It can be seen that the density, viscosity and water content are in the range of typical values for tars derived from biomasses similar to those in this work. For example, Hasanah et al. [82] observed between 7 and 24 %wt. water in coconut shell tar, which had densities between 0.99 and 1.13 g $mL^{-1}$; Zhang et al. [83] reported 21 %wt. water in rice husk tar, separated

from the aqueous phase by centrifugation. Weerachanchai et al. [84] observed densities in the 1.0–1.2 g cm$^{-3}$ range for tar from cassava pulp residue, palm shell and palm kernel obtained by slow pyrolysis at temperatures between 400 and 700 °C.

**Table 3.** Tar composition and properties (%wt.).

|  | RH | ZW | PW |
|---|---|---|---|
| Water | 12.1 | 11.3 | 8.5 |
| Ultimate analysis (dry basis) |  |  |  |
| C | 62.6 | 61.5 | 60.3 |
| H | 6.3 | 6.4 | 6.8 |
| O | 28.8 | 30.7 | 31.6 |
| N | 2.3 | 1.4 | 1.3 |
| Higher heating value (HHV, MJ kg$^{-1}$) | 28.0 | 27.2 | 26.5 |
| Density, 25 °C (g mL$^{-1}$) | 1.01 | 0.98 | 0.97 |

Tars show high contents of carbon and oxygen, which are in the range of reports from the literature. The amount of nitrogen in the tar from rice husk was higher than in that derived from woods, as a consequence of the higher concentrations of proteins. Also, for rice husk, the amount of carbon was slightly higher, probably due to the higher content of hemicellulose, according to a report by Muley et al. [69]. Horne and Williams [85] reported values of approximately 35 %wt. O, 6 %wt. H and 59 %wt. C for tar obtained from a mixture of waste woods. Scholze and Meier [42] observed between 58 and 70 %wt. C, between 5 and 7 %wt. H, between 26 and 35 %wt. O and less than 1.5 %wt. N in tars derived from the pyrolysis of different softwoods, hardwoods and cereal shells.

### 3.5. Gas Chromatography–Mass Spectrometry (GC-MS) of Bio-Oil

Gas chromatography is commonly used to determine the composition of aqueous phases in pyrolysis liquids [19,86]. This technique allowed us to identify and quantify most of the organic compounds in the aqueous phases in the liquid products from the three biomasses and all the components in tars with an $m/z$ ratio lower than 600.

Table 4 shows the compositions and some properties of the aqueous phases obtained in the pyrolysis process. The content of water was very high in all cases and consistent with other reports in the literature for wood sawdusts, cereal shells and other biomass constituents; for example, 84.4 %wt. for wheat shell [87]; 85 %wt. for soybean shell [88]; about 45 %wt. for pine sawdust, and 60 %wt. for cellulose and 75 %wt. for lignin isolated from pine sawdust [69]; and between 48 and 56 %wt. for cassava pulp residue, palm shell and palm kernel [84].

**Table 4.** Composition (%wt.) and properties of aqueous phases in pyrolysis.

|  | RH | ZW | PW |
|---|---|---|---|
| Chemical composition |  |  |  |
| Acids | 38.5 | 43.4 | 32.5 |
| Esters | 0.7 | 0.3 | 0.5 |
| Aldehydes | 2.3 | 1.8 | 2.0 |
| Ketones | 15.1 | 18.9 | 18.1 |
| Furans and heterocyclic compounds | 9.9 | 8.0 | 9.2 |
| Alcohols | 4.2 | 4.4 | 3.0 |
| Phenols | 9.3 | 3.5 | 3.8 |
| Phenolic ethers | 7.1 | 5.5 | 7.4 |
| Unknown | 12.9 | 14.3 | 23.6 |
| Properties |  |  |  |
| Water | 92.6 | 74.0 | 85.2 |
| Density, 25 °C (g mL$^{-1}$) | 1.016 | 1.013 | 1.019 |
| pH | 4 | 4 | 4 |

Acetic acid was a major organic component in all the cases, representing about 90% of all the acids; this compound originates from the cracking of acetyl groups linking xyloses in hemicellulose, which can be found in large quantities in the three raw biomasses tested (see description in Section 3.1.1).

The most important aldehydes were cinnamaldehyde and octanal; the ketones included linear (mainly 2-pentanone, 2,3-pentadione and acetone) and five- and six-carbon-atom ring cyclic representatives (mainly 2-cyclohexen-1-one and 2-hydroxy-3-methyl-2-cyclopenten-1-one). The esters included methyl, ethyl and propyl acetates and propyl butanoate. The major furans were 5-(hydroxymethyl) furfural (between 45 and 55% of total furans), furfural (between 25 and 35%) and 2-furanmethanol.

Methanol, which, according to various sources, is produced from the fragmentation of methoxy groups substituting glucuronic acid in hemicellulose [89] and from the decomposition of cellulose [90], was the only alcohol observed in the three aqueous fractions in liquid products.

Phenol was the most important phenolic compound, representing more than 55% in the group in the aqueous phase from rice husk and about 40% in the cases of wood sawdusts. Following a decreasing order of importance in the group, methylphenols, dimethylphenols and 4-ethyl-1,3-benzenediol can be mentioned, with rice husk being the raw biomass which produces them most abundantly. Guaiacol, syringol, methylguaiacols, methoxycatechol, 4-propylguiacol and 4-ethylguaiacol were the phenolic ethers produced most significantly in all cases.

Other authors reported the same compounds and similar distributions in bio-oil from rice husk [91,92] and pine sawdust [69].

Table 5 shows the compositions of the tars from the different raw biomasses, as determined from the relative areas in the chromatograms. A total of 140 compounds were confidently identified, which were gathered into 11 groups, as shown in Table 5. The FTIR (Section 3.6) and NMR (Section 3.7) analyses show the occurrence of the majority of aromatic compounds. In terms of chromatographic areas, aromatic compounds represented 66.8% in Tar RH, 80.0% in Tar ZW and 80.8% in Tar PW.

**Table 5.** Group composition (%area) of tar phases in pyrolysis.

|  | **RH** | **ZW** | **PW** |
|---|---|---|---|
| Chemical composition |  |  |  |
| Acids | 4.2 | 2.3 | 2.4 |
| Aldehydes | 0.4 | 0.0 | 0.0 |
| Ketones | 10.3 | 6.7 | 5.9 |
| Esters | 1.3 | 0.5 | 0.6 |
| Ethers | 1.2 | 5.8 | 5.3 |
| Furans | 11.4 | 2.9 | 3.4 |
| Pyrans | 0.8 | 0.5 | 0.3 |
| Alcohols and sugars | 5.0 | 2.6 | 4.7 |
| Phenols | 52.6 | 60.7 | 60.7 |
| Hydrocarbons | 0.1 | 8.3 | 8.8 |
| Others | 1.9 | 0.0 | 0.8 |
| Unknown | 10.8 | 9.7 | 7.3 |

The complete list of identified components and their contributions in each tar is shown in Table S1 in the Supplementary Materials.

The number of acids in all the tars was much smaller than in the corresponding aqueous phases, with acetic acid being the most important one in the group.

Among the sugars, 1,6-Anhidro-β-D-glucopiranose (levoglucosan), from the degradation of plant carbohydrates [56] such as sugar cane [93], was the most important one found in all the tars, representing 1.62, 1.19 and 2.95% of the total area in Tar RH, Tar ZW and Tar PW, respectively. The occurrence of levoglucosane was also confirmed by FTIR analysis, as shown in Section 3.6. Zhang et al. [94] showed that in pyrolysis products,

levoglucosan is the result of the degradation of cellulose. Li and Zhang [95] found high yields of levoglucosan in the nonaqueous phases of products in the pyrolysis of waste newspaper and waste cotton at 420 °C.

The content of ketones was the highest in Tar RH, among which hydroxyacetone, 1-acetoxy-2-propanone, 1,2-cyclopentanedione, 3-methyl-2-cyclopenten-1-one and 3-methyl-1,2-cyclopentanedione were outstanding. The tars from wood sawdusts showed higher ether contents, particularly aromatics (1,2,3-trimethoxybenzene and 3,4,5-trimethoxytoluene) and hydrocarbons.

All the tars mainly included phenols, which represented about 53% of the total chromatographic area in Tar RH and 60.7% in Tars ZW and PW. Consequently, a sounder analysis was performed on this information, which is shown in Figure 4, where the phenolic compounds were grouped according to their chemical nature into phenols (including their alkylated derivatives), benzenediols (including compounds with two hydroxyl groups attached to the benzenic ring), guaiacols (including guaiacol and its derivatives), syringols (including syringols and their derivatives) and "others" (including phenolic compounds which do not belong to the previous groups). The same classification was defined by Wang et al. [17] to study wood bio-oils.

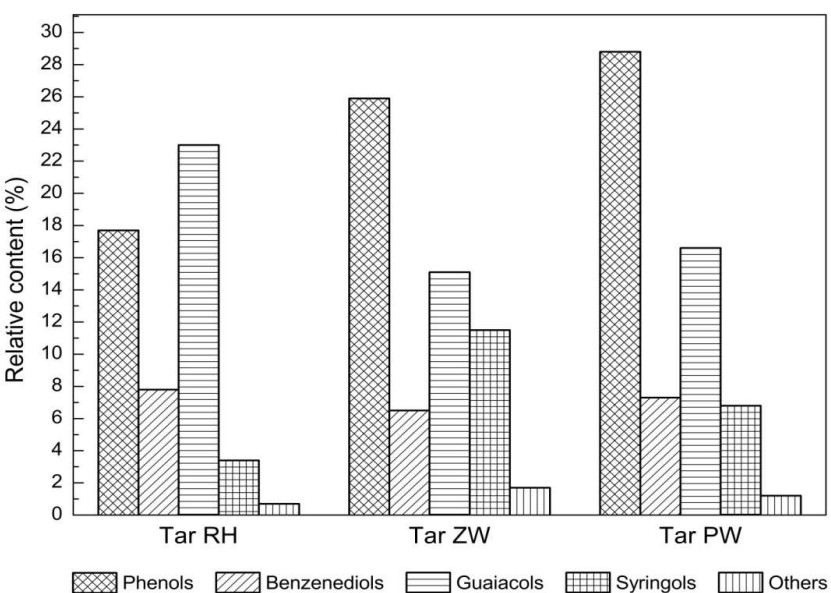

**Figure 4.** Distribution of phenolic compounds in tars from rice husk, zoita wood sawdust and pine wood sawdust.

It can be seen in Figure 4 that Tar RH included a higher content of guaiacols (23.0%) than wood-derived tars, with the most important compound being o-guaiacol. Tars from zoita and pine woods showed high contents of phenols and alkylated phenols, representing 25.9% and 28.8%, respectively, with p-cresol being the most important compound in the group. Syringols, particularly 2,6-dimethoxyphenol, were present in a high concentration (11.5%) in Tar ZW. Benzenediols showed about the same concentration, close to 7% in all the tars, including compounds such as catechol, methylcatechol, hydroquinone and methylhydroquinone, among others.

Phenolic compounds could also be classified according to their source, considering the H, G and S units in lignin [56,96]. This grouping is shown in Figure 5. It should be noted that only those phenolic compounds for each tar whose origin was known for certain were included in each group (H, G, and S); thus, their addition was about 60% in each case. The proportions of phenolic compounds in each tar derived from the H units in lignin, such as phenol, 4-methylphenol and 4-ethylphenol, were similar, while a substantial difference was observed for derivatives from the G and S units. In effect, Tar RH showed the highest proportion of phenolic compounds derived from G units, such as guaiacol, 4-

methylguaiacol, 4-ethylguaiacol, eugenol, and cis and trans isoeugenol. Moreover, Tar ZW and Tar PW showed the highest proportions of derivatives from S units, such as syringol, syringaldehyde, acetosyringone, propiosyringone and 4-allylsyringol.

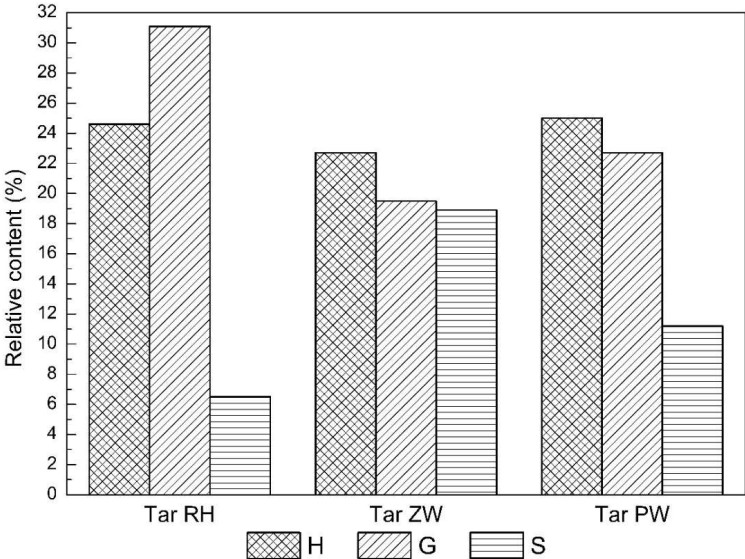

**Figure 5.** Distribution of phenolic compounds in tars from rice husk, zoita wood sawdust and pine wood sawdust according to their source in lignin.

Similar results were shown by other authors. For example, Rosado et al. [56] showed that the pyrolysis of rice husk-isolated lignin mainly produced bio-oils with phenolic compounds derived from G units, with those from H and S units produced to a much lower extent. Analogously, a study by Scholze et al. [96] demonstrated that the pyrolysis of both softwood and hardwood yielded derivatives from S and G units with proportions of 58–79% and 21–41% among the phenolic compounds in tar, respectively.

### 3.6. Fourier Transform Infrared Spectroscopy (FTIR) of Tars

Figure 6 shows the FTIR spectra corresponding to tars from the different raw biomasses, where it can be seen that the main functional groups occurring in the constituents of tars, as indicated by the absorption bands, are coincident in the three cases. It is easier to analyze bands located at wave numbers larger than 1200 cm$^{-1}$, as many vibrations occur in the 1200–700 cm$^{-1}$ range [97]. In order to perform a sounder analysis, the spectra were fitted to Gaussian profile functions using a conventional least squares algorithm, with the conclusions from the fitting exercise shown in Table 6.

**Table 6.** Band assignments in FTIR spectra of tars.

| Wavenumbers (cm$^{-1}$) | Band Assignment |
| --- | --- |
| 3750–3250 | O–H stretching |
| 3000–2850 | Aromatic and aliphatic C–H stretching |
| 1750–1650 | C=O stretching (unconjugated) |
| 1612–1516 | Aromatic skeletal vibration plus C=O stretching |
| 1500–1490 | Aromatic skeletal vibration |
| 1480–1400 | C–H deformations, asymmetry in –CH$_3$ and –CH$_2$– |
| 1400–1330 | Aliphatic C-H and O-H in-plane bending |
| 1325–1250 | C–O of syringyl and guaiacyl rings |
| 1206–1170 | C–C plus C–O plus C=O stretching |
| 1155–1116 | Aromatic C–H deformation in syringyl and guaiacyl rings |
| 1035–980 | Aromatic C–H in-plane bending, plus C–O deformations in primary alcohols, plus C=O stretching (unconjugated) |

**Table 6.** *Cont.*

| Wavenumbers (cm$^{-1}$) | Band Assignment |
| --- | --- |
| 924–870 | Aromatic C–H out-of-plane bending |
| 850–700 | Aromatic C–H bending, furan |
| 680–611 | O–H out-of-plane bending |

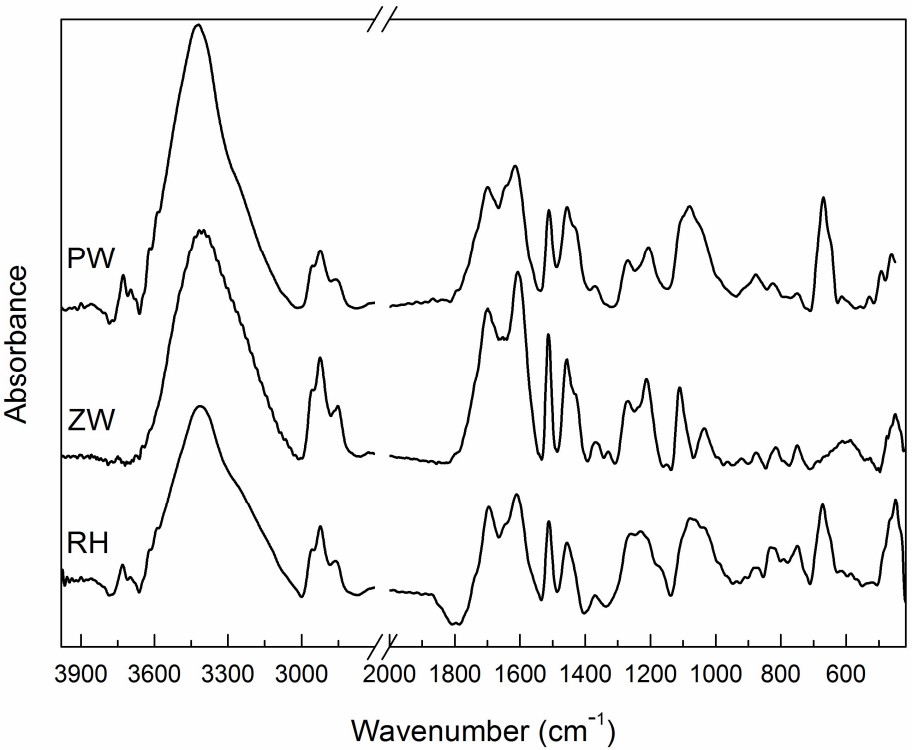

**Figure 6.** FTIR spectra of tars from rice husk, zoita wood sawdust and pine wood sawdust.

The bands at 1750–1650 cm$^{-1}$ are attributed to carbonyl stretching, which indicates the occurrence of ketones [98], aldehydes, esters and carboxylic acids [99]. It can be noticed that the intensity of bands in that wavenumber range in tars from wood sawdusts is double that from rice husk. These bands are consistent with NMR signals in the 1.5 to 3.0 ppm region (see the following section), which show aliphatic protons in $\alpha$ carbon atoms in aromatic rings or in saturation.

Carbonyl groups are particularly important in biomass pyrolysis liquid products, not only for their abundance but also for their reactivity, which impact product stability and limit upgrading or the options for direct use. Figure 7 shows the deconvolution of tar FTIR absorption spectra between 1750 and 1400 cm$^{-1}$, which allows us to identify the contribution from carboxylic acids (band at 1713 cm$^{-1}$) in the group of carbonyl-containing compounds [100]. The tar from rice husk pyrolysis shows a shoulder at 1725 cm$^{-1}$, which could correspond to a more important contribution from ketones [101]. In all cases, the band at 1606 cm$^{-1}$ corresponds to vibrations from aromatic rings presenting many substitutions [100].

The tars from wood sawdusts also showed a band at 1420 and 1350 cm$^{-1}$ produced by the C–H in-plane deformation with aromatic ring stretching [17]. All tars also showed a stronger intensity in bands at 3000–2850 cm$^{-1}$ (C–H stretching) and 1612–1516 cm$^{-1}$ (C=C stretching in the aromatic ring), suggesting high aromaticity [102].

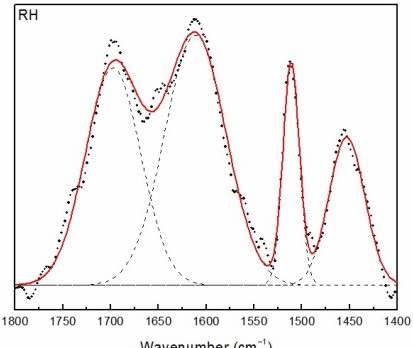 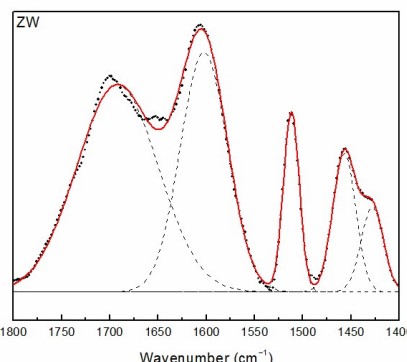 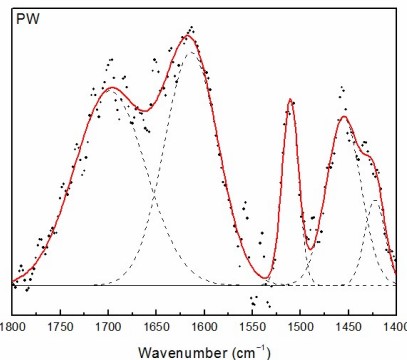

**Figure 7.** Deconvolution of the FT-IR spectra of tars from rice husk, zoita wood sawdust and pine wood sawdust in the 1750–1400 cm$^{-1}$ region.

All the tars showed a band due to C–O bond stretching vibration at 1033 cm$^{-1}$, showing the existence of aliphatic alcohols and aliphatic and alicyclic eight-membered rings, particularly levoglucosan; this compound can be considered an index of the degree of progress of pyrolysis, as it is an intermediate product of the primary pyrolysis reactions of cellulose and hemicellulose [14,103]. The tar from rice husk showed a larger proportion of this band as compared to the other biomasses, a consequence of its higher content of hemicellulose.

Bands between 3750 and 3250 cm$^{-1}$ can be assigned to the stretching of O–H bonds linked to carbon atoms in alcohol groups (aliphatic alcohols and phenols) [96,98], with the contribution from OH groups in carboxylic acids being possible as well [99]. The band at 3750 cm$^{-1}$ shows the existence of water [83], the content of which, for all cases, is shown in Table 4 (Section 3.4). This band was the most important in all the spectra, accounting for more than 50% of the total absorbance. Alcohol groups in biomass pyrolysis are produced from all the three major components in biomass (cellulose, hemicellulose and lignin) [37,54,74]. For example, in the thermal decomposition of cellulose and hemicellulose, the glycosidic bonds break and are replaced by hydroxyl bonds; in the depolymerization of lignin, phenolic compounds are produced which contribute to C–O and O–H vibrations.

The bands between 1400 and 1330 cm$^{-1}$ (OH in-plane bending) and between 680 and 611 cm$^{-1}$ (OH out-of-plane bending) also correspond to the associated O–H groups [98,102], with the signal intensities higher in the case of pine sawdust tar.

Bands from 1325 to 1250 cm$^{-1}$ are characteristic of C–O bonds in syringyl and guaiacyl rings [17]. The tars from wood sawdusts showed higher band intensities in this range, surely indicating more phenolic ethers, such as eugenol, vanillin, vinylguaiacol, methylguaiacol and guaiacol, among others, than rice husk tar. These compounds, with much higher concentrations in wood sawdust tars than in rice husk tar, are due to lignin degradation (see Table 1) [103]. However, tar from rice husk showed a higher intensity at 1274 cm$^{-1}$, corresponding to guaiacyl rings [17], while the signals of syringyl rings were missing at 1329 cm$^{-1}$ [17], consistent with the composition of its lignin (see Section 3.1.1). Complementarily, the intensity of the band at 1116 cm$^{-1}$ (corresponding to aromatic C–H deformation in the syringyl ring [17]) was higher in tars derived from wood sawdusts, while the band at 1155 cm$^{-1}$ corresponding to aromatic C–H in-plane deformation in the guaiacyl ring [17] was more intense in tar from rice husk. This is consistent with the GC-MS observations shown in Figure 5.

The bands in the ranges 924–870 cm$^{-1}$ and 850–700 cm$^{-1}$ are assigned to furfural and furans [101,104], which were intense in tar from rice husk, according to the higher content of hemicellulose in this biomass (refer to Tables 1, 5 and S1).

*3.7. $^1$H Nuclear Magnetic Resonance (NMR) of Tars*

NMR spectra provide complementary information for functional groups observed in FTIR spectra and permit the integration and comparison of their areas [105]. Figure 8

shows the [1]H NMR spectra of the various pyrolytic tars, and the relative areas of selected regions are included in Table 7. The signal from the solvent, which was excluded in the integration, can be observed at 7.26 ppm.

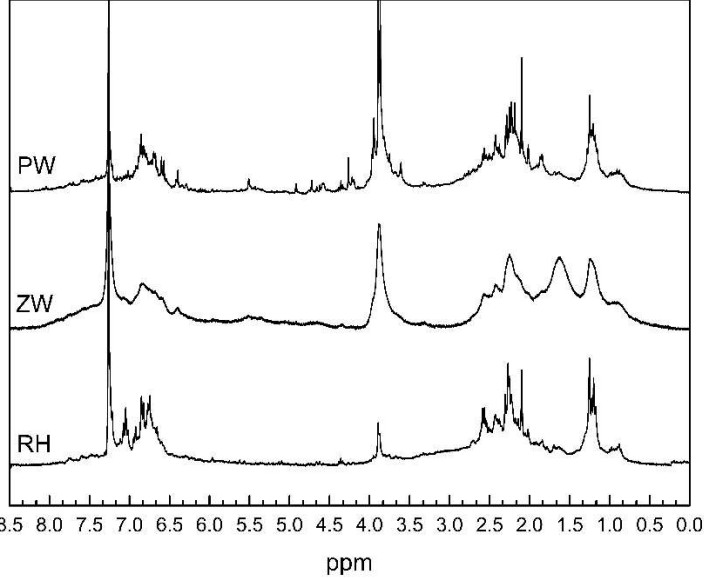

**Figure 8.** [1]H NMR spectra of tars from rice husk, zoita wood sawdust and pine wood sawdust.

The amount of hydrogen taking part of the aliphatic functional groups, mainly the methyl or methylene groups in the β or γ position in an aromatic ring or unsaturation, which are shown to be in the 0.5 to 1.5 ppm range, was similar to the three tars. This evidence pertaining to hydrogen suggests the occurrence of aromatic compounds with aliphatic substitutions (ethyl or heavier) [106] and/or aromatic rings linked by aliphatic bonds, as observed in the GC analysis (see Section 3.5).

**Table 7.** H atom distribution in tars according to [1]H NMR (%area).

| Chemical Shift (ppm) | Assignment | RH | ZW | PW |
|---|---|---|---|---|
| 0.5–1.5 | Aliphatic protons of methyl or methylene groups in β or γ position in an aromatic ring or unsaturation | 15.1 | 15.2 | 15.0 |
| 1.5–3.0 | Aliphatic protons in α position in an aromatic ring, unsaturation or heteroatom | 43.3 | 38.4 | 40.8 |
| 3.0–4.4 | Protons on carbon atoms next to an aliphatic alcohol or ether, or a methylene group that joins two aromatic rings | 10.3 | 11.4 | 21.7 |
| 4.4–6.0 | Protons on carbon atoms in methoxy groups and protons of carbohydrate-like molecules | 1.9 | 6.3 | 1.0 |
| 6.0–8.5 | Protons linked directly to an aromatic ring | 29.4 | 28.6 | 21.5 |

Tar RH showed the highest content of protons in aliphatic carbon atoms linked to aromatic or olefinic C=C bonds or to a heteroatom (in the region from 1.5 to 3.0 ppm in the spectra) and of protons in aromatic rings (in the region from 6.0 to 8.5 ppm in the spectra). This suggests that substitutions in aromatic rings in this tar mainly occur by means of C–C bonds (for example, methyl groups) and that they contain a higher proportion of benzenic rings or fewer substituents in the benzenic rings than the other tars [106].

Tar PW showed twice the number of protons in carbon atoms in aliphatic alcohols or ethers (in the region from 3.0 to 4.4 ppm in the spectra) than the others. Moreover, this tar showed the lowest number of aromatic protons (in the region from 6.0 to 8.5 ppm in the spectra), thus suggesting that there are fewer aromatic rings in the tar or that the rings are highly substituted; according to the FTIR observations, the second option seems more feasible.

Protons in aromatic ethers (in the region from 4.4 to 6.0 ppm in the spectra), mainly corresponding to methoxyphenols derived from lignin, such as guaiacol, syringol and their derivatives [106,107], were much more numerous in Tar ZW than in the others, as confirmed by the GC-MS analysis and also observed in the FTIR analysis. This region in the spectra could also correspond to protons in molecules similar to carbohydrates; thus, in the case of Tar RH, it could represent molecules such as levoglucosan, consistent with the higher band intensity for this compound in the FTIR spectrum (see Section 3.6) and GC-MS analysis (see Table S1).

### 3.8. Molecular Weight Range of Tars

Pyrolytic tars include compounds with a wide distribution of molecular weights, such as polyaromatic hydrocarbons and phenols substituted with different lengths and branching chains, among others. Size exclusion chromatography was used to determine the distribution of molecular weights in tars and to provide a view of their polymeric nature. Figure 9 shows the distributions, which are wide in all cases.

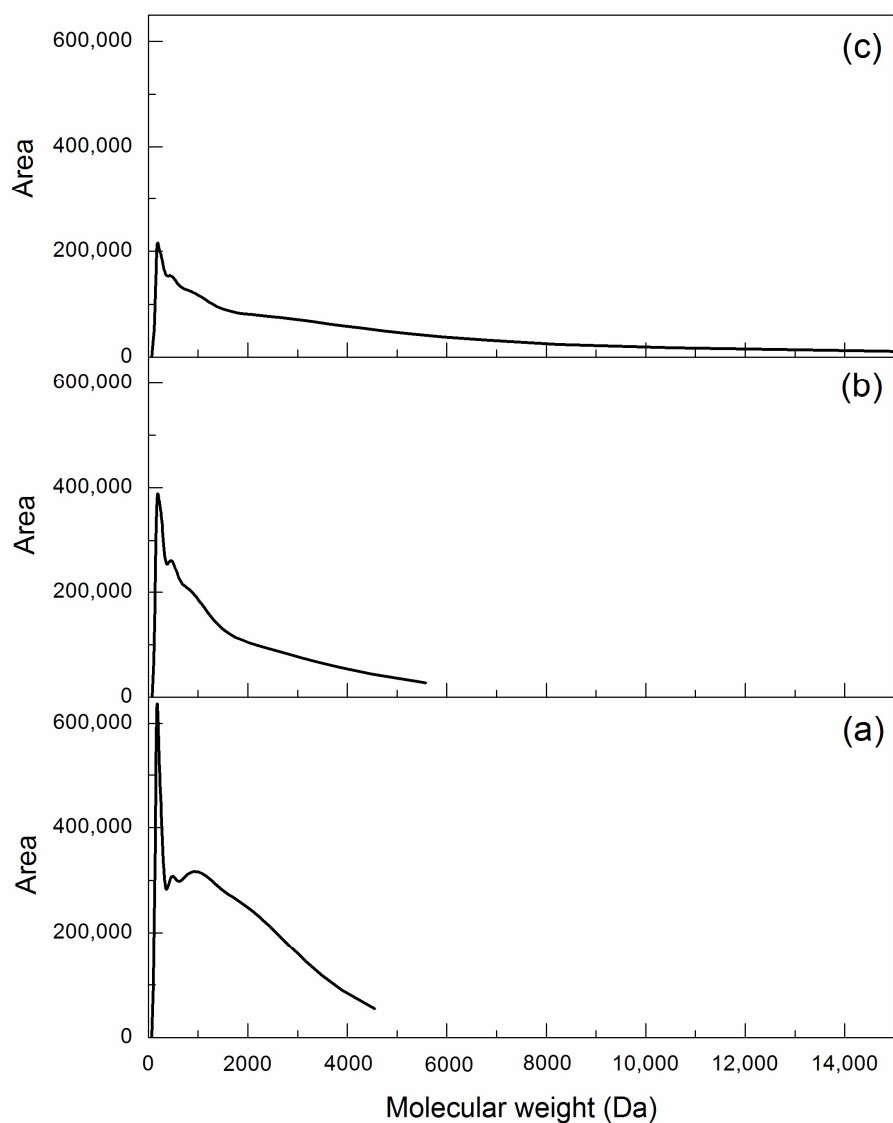

**Figure 9.** Molecular weight distribution in tars. (**a**) Rice husk; (**b**) zoita wood sawdust; (**c**) pine wood sawdust.

The chromatograms and the calibration of molar mass (MM) provide the information necessary to calculate the average molecular weight number (Mn), the average molecular

weight (Mw), the peak molar mass (Mp) and the dispersity D = Mw/Mn. The last parameter provides a description of molecular weight heterogeneity in the sample, that is, broadness in the distribution curve. Table 8 shows the results based on the standard polystyrene, where it can be seen that Mw and D in the case of Tar PW were much larger than in Tar ZW and Tar RH, with Mn and Mp being similar in all cases. The information from Mw and D suggests that Tar PW includes high branching in its chains, with poorly ordered growth during pyrolysis reactions, probably as a consequence of the higher recombination speed of lignin units during pyrolysis [106]. Fahmi et al. [108] proposed a correlation to demonstrate that higher lignin content in a biomass determines higher Mw values in the corresponding pyrolytic bio-oil; consistent with this, PW showed the highest lignin content in this study (see Table 1).

**Table 8.** Molecular properties of tars.

|  | **RH** | **ZW** | **PW** |
|---|---|---|---|
| Average molecular weight number (Mn) (g mol$^{-1}$) | 327 | 322 | 385 |
| Average molecular weight (Mw) (g mol$^{-1}$) | 855 | 809 | 5643 |
| Dispersity (D = Mw/Mn) | 2.6 | 2.5 | 14.7 |
| Peak molar mass (Mp) (g mol$^{-1}$) | 178 | 190 | 188 |

Similar results were reported by Prauchner et al. [109] upon characterizing tar through the slow pyrolysis of eucalyptus wood at 500 °C and 14 °C h$^{-1}$, determining a Mw of 2100 g mol$^{-1}$; after the thermal treatment of tar at 250 °C for 2 to 8 h, the Mw increased to 3100–6200 g mol$^{-1}$.

Scholze et al. [96] and Bayerbach et al. [110] determined that the Mw of pyrolytic tars derived from various wood sawdusts (beech, eucalyptus, pine and poplar) were from 650 to 1300 g mol$^{-1}$ in the cases of the last three, suggesting that their structures mainly corresponded to trimers and tetramers of the lignin units G, H and S. On the contrary, the much higher Mw of tar from beech sawdust, about 10,000 g mol$^{-1}$, was attributed to less extended lignin cracking during pyrolysis. Mullen and Boateng [106] reported Mws from 2566 to 6896 g mol$^{-1}$ in pyrolytic tars from oak wood, barley straw and hull, switchgrass and soy straw, determining that most of the components in tars, about 30%, corresponded to phenol dimers and trimers.

Espinosa-Acosta et al. [111] reported that the Mw ranges for organosolv and ionic lignin were 500–4000 g mol$^{-1}$ and 2220–6347 g mol$^{-1}$, respectively. As pyrolytic tars from these raw materials showed similar Mw values, it is possible to infer that their polymeric characteristics were similar, thus suggesting that they could be appropriate replacements for lignin in various applications.

*3.9. Possible Applications of Tars*

The three pyrolytic tars were shown to be a complex mixture of high-molecular-weight oxygenated compounds and hydrocarbons, as expected, with smaller amounts of water and acids than their corresponding aqueous phases in bio-oils. The evidence shown in the previous sections supports the possibility of their use for various applications.

The high content of hydrocarbons in tars from woods suggests that they could be co-processed in FCC units to produce fuels and petrochemical raw materials. This approach has been suggested to valorize resources from biomass, including pyrolytic and vegetable oils and animal fats, based on the fact that FCC commercial catalysts (Y zeolite supported on a mesoporous matrix) are appropriate to convert oxygenated compounds into hydrocarbons without altering the usual refinery operation [106,112]. Tars are more suitable to be co-processed, as the high water content in aqueous fractions impedes them from being incorporated with standard feedstocks over a certain maximum concentration, given by the operative conditions in [23].

It was shown in Section 3.5 that tars include a high concentration of phenolic compounds, derived from the lignin fraction in biomass, which are the most important group.

Moreover, the tars exhibited a polymeric nature, thus suggesting that they could be used to partially replace phenol in the production of phenol-formaldehyde resins [113,114]. This characteristic can be observed in tars from many different raw biomasses, such as hard and soft woods, sugar cane bagasse, cereal shells and lignin from paper pulping. Phenolic compounds in tars, on average, are less reactive in polymerizing than phenol itself [115], but tars can substitute up to 50% of phenol in the production of resins [116].

Tars could also be incorporated with asphaltic ligands, partially replacing fossil resources, which could be redirected to higher-value products [117]. There exists a high worldwide demand for asphalts as a consequence of the need for both new roads and repaving, as asphalts suffer from chemical changes, such as modifications in their colloidal structure and oxidation, leading to the loss of the required physical properties. In mixtures, tars could also act as antioxidants extending their useful life [118,119].

Tars could also be repolymerized in the production of carbon fibers [120], or they could replace polymers in the obtention of microspheres as supports for biocides and agrochemicals, with slow release of the active components. In the second application, the polymeric characteristics, composition, density and viscosity of tars could make them possible precursors of matrices to encapsulate active principles. Taverna et al. [121] studied new films based on tar microparticles loaded with sodium alginate and eugenol, and their mechanical properties and antibacterial activity were adequate. They also prepared films with different reticulation degrees using $CaCl_2$ and assessed the controlled release of eugenol.

There exist concerns about producing carbon fibers from polyacrylonitrile, a process which releases toxic gases. To solve this, lignin from biomass is considered an alternative precursor, but the resulting mechanical properties of the fibers are poor. Pyrolytic lignins (tars) would perform better than crude lignin from biomass in catalytic repolymerization to produce carbon fiber precursors, as their phenolic units, which have reactive functional groups such as hydroxyl, carbonyl and vinyl, are smaller [120].

Moreover, it was postulated that tars could be a source for the separation of various chemicals, such as hexamethylbenzene [122] and several other aromatic hydrocarbons and phenols [123,124].

## 4. Conclusions

Wood sawdusts yield more bio-oil and gases than rice husk in pyrolysis at 500 °C. In turn, rice husk yields more char, consistent with its high content of ash. Moreover, tars represent a much higher fraction of bio-oils in the case of wood sawdusts. As most of the components of tars are derived from lignin, these observations can be associated with the higher content of lignin in raw wood biomasses.

Carbon is more concentrated and oxygen is less concentrated in all the tars as compared to the raw biomasses, thus indicating a neat concentration of energy in the pyrolysis product.

All the tars include important amounts of phenols, representing about 53.0% of the total chromatographic area in Tar RH and 60.7% in Tars ZW and PW. However, their distributions are different according to the source biomass: guaiacols prevail in Tar RH, while phenol and alkylated phenols predominate among wood tars.

The combination of results from the [1]H NMR, FTIR and GC-MS analyses confirm that Tar RH contains benzenic rings with smaller-sized substituents than the wood tars. In turn, Tar PW shows highly substituted benzenic rings. Protons in aromatic ethers mainly corresponding to methoxyphenols derived from lignin, such as guaiacol, syringol and their derivatives, are much more numerous in Tar ZW than in the other tars, as confirmed by the GC-MS and FTIR analyses.

As deduced from the SEC analysis, tars exhibit polymeric characteristics which are comparable with those of organosolv lignin. Moreover, the content of lignin in the source biomass determines the average molecular weight of the tar pyrolysis product.

The extensive characterization of tars provided useful information to back the possible applications of these liquid products. Among others, these could include the replacement

of polymers to produce microspheres supporting slow-release biocides and agrochemicals, or of asphalt binders in road paving; the partial replacement of phenols in the formulation of phenol-formaldehyde resins; their use as raw materials to produce carbon fibers through repolymerization; and, at a larger scale, FCC co-processing together with standard fossil feedstocks.

**Supplementary Materials:** The following supporting information can be downloaded at: https://www.mdpi.com/article/10.3390/pr12040817/s1, Table S1: Composition of pyrolysis tars determined by GC-MS (%area).

**Author Contributions:** Conceptualization, P.S. and M.B.; Methodology, P.S., C.A.B., U.S., M.F. and R.P.; Formal Analysis, P.S., M.B. and U.S.; Investigation, P.S., C.A.B. and R.P.; Resources, M.F. and U.S.; Data Curation, P.S. and M.B.; Writing—Original Draft Preparation, M.B. and P.S.; Writing—Review and Editing, U.S.; Project Administration, M.F.; Funding Acquisition, M.F. and U.S. All authors have read and agreed to the published version of the manuscript.

**Funding:** This work was carried out with financial support of the Universidad Nacional del Litoral (UNL, Santa Fe, Argentina), Proj. CAID 50620190100177LI; the Consejo Nacional de Investigaciones Científicas y Técnicas (CONICET), PIP 2021-3146; and the Agencia Nacional de Promoción Científica y Tecnológica (ANPCyT), PICT 2019/3391 and 2019/2621.

**Data Availability Statement:** The data are contained within the article.

**Conflicts of Interest:** The authors declare no conflicts of interest.

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
