# Peer review of "Characterization of Pyrolytic Tars Derived from Different Biomasses"

_processes, doi:10.3390/pr12040817_

Round 1
Reviewer 1 Report
Comments and Suggestions for Authors
This manuscript presents research that is interesting and important from the point of view of producing tar from biomass in the pyrolysis process. However, before the work is accepted for publication, it is advisable to improve the following elements:
1. The authors use different names for the reactor: "fixed bed, multipurpose unit at the pilot scale" in abstract and in section 2.2 "downdraft reactor". Do the authors believe that a 120 cm long reactor can be called a pilot scale?
2. Avoid lumping references as in [4,5], [9,10], [16,17], [18,19], [22,23], [24,25] and all other. Instead, summarise the main contribution of each referenced paper in a separate sentence.
3. Strengthen the justification for conducting the biomass pyrolysis process. Completion of the Introduction with how much heat is required for the pyrolysis process of various types of biomass.
4. What impact could different fraction sizes of raw materials have on yields and product quality? I mean, RH (1.7 - 2.4 mm) and PW , ZW (1.7 - 4.7 mm).
5. The sample size used in the study may not be representative of all types of biomass, which limits the generalisability of the results.
6. Section 2.2. Pyrolysis experiment. This section requires clarification:
a) "Solid products (char, ash)" - is ash produced during pyrolysis?
b) "Ignition in the reactor was produced by the combustion of propane" - Is the reactor heated by combustion of propane?
c) "Reaction temperature in the experiments was 500 ºC" - how and at what point was the temperature measured? Was the temperature uniform along the length of the reactor?
d) It would be good if Figure 1 represented a pyrolysis reactor. Currently, it contains air inlets that are probably not used during pyrolysis.
7. The notation after Table 1 "a=100-ash" is misleading. The letter "a" should be next to oxygen (in ultimate analysis) and next to other (component analysis).
8. In Figure 2a), does the sample weight increase above 800 oC? Explain what caused this increase?
9. It is understood that individual components may signal different ion currents. However, I assume that Figures 3a-c, e.g. for acetic acid, were prepared for one m/z value (43, 45, 60). Which? You may specify this for each chemical component.
10. "the heating value being 7135 kcal kg-1." When referring to values from the literature, they should be converted to units used in the work (kJ/kg).
11. What are the limitations of a downdraft reactor from a commercial point of view?
12. Conclusion. "At a larger scale, FCC co-processing together with standard fossil feedstocks is demonstrated." Were such studies conducted in this work?
Author Response
REVIEWER 1
This manuscript presents research that is interesting and important from the point of view of producing tar from biomass in the pyrolysis process. However, before the work is accepted for publication, it is advisable to improve the following elements:
1.The authors use different names for the reactor: "fixed bed, multipurpose unit at the pilot scale "in abstract and in section 2.2 "downdraft reactor". Do the authors believe that a 120 cm long reactor can be called a pilot scale?
Following the suggestion by the reviewer, we decided to homogenize the denomination of the reactor used in the experiments by adopting the more common designation “bench scale” (see changes in lines 9 and 156).
- Avoid lumping references as in [4,5], [9,10], [16,17], [18,19], [22,23], [24,25] and all other. Instead, summarise the main contribution of each referenced paper in a separate sentence.
Following the suggestion by the reviewer, we decided to rewrite an important number of sentences along the manuscript (one per reference) and selected the most relevant paper when it was not convenient to write a separate sentence with the most important contribution from each reference. This happened particularly in cases where the same reference was used more than once. Changes are shown highlighted in the manuscript (see lines 49, 61, 91, 97, 99, 103, 105, 238, 244, 252, 305, 328 to 329, 338 to 339, 429 to 431, 443, 446, 468, 552, 567, 640, 663, 814)
3.Strengthen the justification for conducting the biomass pyrolysis process. Completion of the Introduction with how much heat is required for the pyrolysis process of various types of biomass.
In order to address the suggestion by the reviewer, we have now incorporated additional information in the section Introduction in relation to the energy demands to perform the pyrolysis of various biomasses (see lines 76 to 78, and 84 to 88).
4) What impact could different fraction sizes of raw materials have on yields and product quality? I mean, RH (1.7- 2.4 mm) and PW, ZW (1.7 - 4.7 mm)
Particle size is an important issue in pyrolysis processes as, in general, the larger the particles, the more important the restrictions to heat transfer. For example, Shen et al. [1] observed that the yield of bio-oil increased when the particle size of mallee wood sawdust decreased, and Yang et al. [2], who studied different conditions in the pyrolysis of three residual biomasses with very different particle size in a fluidized bed reactor, explained the observed differences in the depolymerization and cracking temperatures based on the changes in the heat conduction due to the different particle sizes.
However, this issue, at least for the particle size range used in this work, will not mask the experimental observations. Previously, by using soybean shell and pine sawdust with particle sizes similar to the ones in this work [3], we had shown that differences in pyrolysis yields and product compositions were due to the different compositions of the raw biomasses.
[1] Shen J., Wang X., Garcia-Perez M., Mourant D., Rhodes M., Li C. (2009) Effects of particle size on the fast pyrolysis of oil mallee woody biomass. Fuel 88, 1810-1817.
[2] Yang S. Wu M., Wu C. (2014) Application of biomass fast pyrolysis part I: Pyrolysis characteristics and products. Energy 66, 162-171.
[3] Bertero M.; Sedran U (2016) Immediate catalytic upgrading of soybean shell bio-oil. Energy 94 (2016) 171-179.
5) The sample size used in the study may not be representative of all types of biomass, which limits the generalizability of the results.
As previously mentioned, particle size is an important operative parameter in pyrolysis processes. However, other parameters being the same, it is the composition of the biomass the main responsible for the resulting product yields and distribution. Even though the particle sizes used in this work cannot be generalized to every biomass, it is to be noted that it is useful to gather information from experiments using the feedstocks (residual biomass) in the same conditions as they are produced, that is, without further processing such as milling and drying.
- Section 2.2. Pyrolysis experiment. This section requires clarification:
- a) "Solid products (char, ash)" - is ash produced during pyrolysis?
The solid product in pyrolysis, named char, is composed by organic material (mainly carbon) and ash. It had been written in lines 69 and 70 of the original manuscript that solid products were “(char, basically formed by carbon and inorganic material)”. We believe this is a proper definition and maintained it, also in line 164.
- b) "Ignition in the reactor was produced by the combustion of propane" -Is the reactor heated by combustion of propane?
The reactor is not heated by the combustion of propane. It is only used to ignite biomass and then the process proceeds autothermally. This was clearly written in lines 149 to 150 of the original manuscript.
- c) "Reaction temperature in the experiments was 500 ºC" - how and at what point was the temperature measured? Was the temperature uniform along the length of the reactor?
The reaction temperature was defined to be that observed at the reactor basis (T1 in Figure 1). However, there existed temperature variations along the bed, which is a characteristic of these units. We have now added the description reactor temperature (see line 168).
- d) It would be good if Figure 1 represented a pyrolysis reactor. Currently, it contains air inlets that are probably not used during pyrolysis.
The reason for the air inlet is that the reactor employed in the experiments can be used in both pyrolysis and gasification processes. However, pyrolysis can be still performed with minor amounts of oxygen in the system. Indeed, in studies devoted to biomass thermochemical processes, pyrolysis can be defined as the processes where the equivalence ratio (ER), that is, the relationship between the actual amount of air used and the stoichiometric amount for complete combustion, is lower than 0.2 [4, 5]. In our work, ER was 0.11, thus obeying this definition. We had stated in the manuscript that the flow of nitrogen was 12.8 L min-1. For the sake of clarity, we have now written this in terms of the flow of air, which was 16 L min-1 (see line 168).
[4] Li, B.; Song, M.; Xie, X.; Wei, J.; Xu, D.; Ding, K.; Huang, Y.; Zhang, S.; Hu, X.; Zhang, S. & Liu, D. (2023). Oxidative fast pyrolysis of biomass in a quartz tube fluidized bed reactor: Effect of oxygen equivalence ratio. Energy 270, 126987.
[5] Polin, J. P., Carr, H. D., Whitmer, L. E., Smith, R. G., & Brown, R. C. (2019). Conventional and autothermal pyrolysis of corn stover: Overcoming the processing challenges of high-ash agricultural residues. Journal of analytical and applied pyrolysis 143, 104679.
7.The notation after Table 1 "a=100-ash" is misleading. The letter "a" should be next to oxygen (in ultimate analysis) and next to other (component analysis).
The title of Table 1 was corrected accordingly.
- In Figure 2a), does the sample weight increase above 800 °C? Explain what caused this increase?
We appreciate the observation by the reviewer. The original data were revised and we noticed there was a slight variation. Figure 2a was now modified.
9.It is understood that individual components may signal different ion currents. However, I assume that Figures 3a-c, e.g. for acetic acid, were prepared for one m/z value (43, 45, 60). Which? You may specify this for each chemical component.
Following the suggestion by the reviewer, the signals used to identify the various compounds were now incorporated to the text of the manuscript (see lines 363 to 364).
- "the heating value being 7135 kcal kg-1." When referring to values from the literature, they should be converted to units used in the work (kJ/kg).
The text was corrected accordingly (see line 483).
- What are the limitations of a downdraft reactor from a commercial point of view?
Downdraft reactors are the preferred option for gasification reactors. However, in order to address the reviewer´s concern, we could mention some limitations. For example, if the bulk density of the feedstock is low, grate blocking, bridging and channeling may occur. Moreover, high humidity could impose low efficiency in the reactor operation.
- Conclusion. "At a larger scale, FCC co-processing together with standard fossil feedstocks is demonstrated." Were such studies conducted in this work?
Those studies were not conducted in the work described in the manuscript. In order to make the text clearer, we have now rewritten this paragraph (see lines 860 to 861).
We would like to mention that our group is strongly devoted to the issue of co-processing pyrolysis products in refineries, particularly in the FCC process, as shown by the following publications.
- Melisa Panero, Richard Pujro, Marisa Falco, Ulises Sedran, Javier Bilbao, José M. Arandes (2022). Limitations in the energy balance when VGO/aqueous bio-oil mixtures are co-processed in FCC units. Fuel 324, 124798.
- Melisa Bertero, Ulises Sedran (2015). Chapter 13: “Co-processing of bio-oil in Fluid Catalytic Cracking” in “Recent Advances in Thermo-chemical Conversion of Biomass”, Ashok Pandey, Thallada Bhaskar y Michael Stocker (Eds). Elsevier: Amstendam. p. 349-375. DOI: http://dx.doi.org/10.1016/B978-0-444-63289-0.00013-2
- Melisa Bertero, Marisa Falco, Ulises Sedran (2014). Chapter 8: “Application of bio-oil derived from plant biomass in petrochemical industry” in “Biotechnology development in Agriculture, Industry & Health. Advanced Conversion Technologies for Lignocellulosic Biomass. Vol. 3. Ani Idris, Ulises Sedran, Zainul Zakaria (Eds), Penerbit UTM Press, Malasia, p. 177-220
- Melisa Bertero, Juan Rafael García, Marisa Falco, Ulises Sedran (2022) FCC matrix components and their combination with Y Zeolite to enhance the deoxygenation of bio-oils. BioEnergy Research 15, 1327-1341. DOI: https://doi.org/10.1007/s12155-021-10322-z
- Richard Pujro, Juan Rafael García, Melisa Bertero, Marisa Falco, Ulises Sedran (2021) Review on Reaction Pathways in the Catalytic Upgrading of Biomass Pyrolysis Liquids. Energy Fuels 35, 16943−16964. DOI: https://doi.org/10.1021/acs.energyfuels.1c01931
- Melisa Bertero, Juan Rafael García, Marisa Falco, Ulises Sedran (2019). Conversion of cow manure pyrolytic tar under FCC conditions over modified equilibrium catalysts. Waste and Biomass Valorization 11, 2925–2933. DOI: https://doi.org/10.1007/s12649-019-00588-y
- Richard Pujro, Melisa Panero, Melisa Bertero, Ulises Sedran, Marisa Falco (2019) Hydrogen transfer between hydrocarbons and oxygenated compounds in co-processing bio-oils in FCC. Energy & Fuels 33, 6473-6482. DOI: https://doi.org/10.1021/acs.energyfuels.9b01133
- Melisa Bertero, Juan Rafael García, Marisa Falco, Ulises Sedran (2019) Equilibrium FCC catalysts to improve liquid products from biomass pyrolysis. Renewable Energy 132, 11-18. DOI: https://doi.org/10.1016/j.renene.2018.07.086
- Melisa Bertero, Juan Rafael García, Marisa Falco, Ulises Sedran (2017) Hydrocarbons from bio-oils. Performance of the matrix in FCC catalysts in the immediate catalytic upgrading of different raw bio-oils. Waste & Biomass Valorization 8 (3), 933-948. DOI: https://doi.org/10.1007/s12649-016-9624-z
- Melisa Bertero, Ulises Sedran (2013) Conversion of pine sawdust bio-oil (raw and thermally processed) over equilibrium FCC catalysts. Bioresource Technology 135, 644-651. DOI: https://doi.org/10.1016/j.biortech.2012.11.070
- Melisa Bertero, Ulises Sedran (2013) Upgrading of bio-oils over equilibrium FCC catalysts. Contribution from alcohols, phenols and aromatic ethers. Catalysis Today 212 (1), 10-15. DOI: https://doi.org/10.1016/j.cattod.2013.03.016
- Melisa Bertero, Gabriela de la Puente, Ulises Sedran (2013) Products and coke from the conversion of bio-oil acids, esters, aldehydes and ketones over equilibrium FCC catalysts. Renewable Energy 60, 349-354. DOI: https://doi.org/10.1016/j.renene.2013.04.017
- Melisa Bertero, Gabriela de la Puente, Ulises Sedran (2012) Fuels from bio-oils. Bio-oil production from different sources, characterization and thermal conditioning. Fuel 95, 263–271. DOI: https://doi.org/10.1016/j.fuel.2011.08.041

Reviewer 2 Report
Comments and Suggestions for Authors
The manuscript investigates the pyrolysis of three different biomasses, namely rice husk (RH), zoita wood sawdust (ZW), and pine wood sawdust (PW), with a focus on the yields and compositions of the liquid products, particularly the tar fractions. Here are some comments and suggestions to enhance the clarity and scientific rigor of the manuscript:
1) The abstract should briefly introduce the issues and synthesize the “message” of your findings to place the abstract in context. That would help prospective abstract readers decide whether to read the entire article.
2) The introduction section should contain a critical analysis of previous literature for what has been done, research gaps, and limitations to justify the novelty of your work.
3) The discussion about the application of the biomass seems to be lacking.
4) It would be more interesting if the authors focused more on the significance of their findings regarding the importance of the interrelationship between the obtained results and sustainable development/cleaner production in the sector context, and the barriers of doing it, what would be the consequences, in the real world, in changing the observed situation, what would be the ways, in the real world, to change/improve the observed situation.
Author Response
The manuscript investigates the pyrolysis of three different biomasses, namely rice husk (RH), zoita wood sawdust (ZW), and pine wood sawdust (PW), with a focus on the yields and compositions of the liquid products, particularly the tar fractions. Here are some comments and suggestions to enhance the clarity and scientific rigor of the manuscript:
1)The abstract should briefly introduce the issues and synthesize the “message” of your findings to place the abstract in context. That would help prospective abstract readers decide whether to read the entire article.
The abstract includes the objectives of our research, a synthetic description of the methodology and the analytical techniques used, as well as the main findings and their possible implications. We have now included a better description of the pyrolysis reactor used.
2)The introduction section should contain a critical analysis of previous literature for what has been done, research gaps, and limitations to justify the novelty of your work.
The introduction of our manuscript addresses the following topics: the use of biomass and lignocellulosic biomass to produce fuels and chemicals, a general overview on biomass pyrolysis, an analysis of the most important previous publications on pyrolysis of the biomasses we used in our manuscript and definitions about liquid pyrolysis products and their possible upgrading.
The novelty of our research and the niche it will hopefully fill are clearly described in the eleventh paragraph in the introduction (new version of the manuscript. We have now extended the introduction section, including references to a technological-economical analysis of the production of transportation fuels by means of thermochemical processes and to the energy requirements of the pyrolysis processes. We have also clarified the objectives of our research.
We would also like to mention that the critical analysis of the state of the art of this subject matter was shown in the introduction section. Moreover, other statements in the previous literature were discussed in the Results section.
3)The discussion about the application of the biomass seems to be lacking.
The objective of our manuscript was not to discuss the applications of the biomass but to focus on the exhaustive characterization of the tars formed in the pyrolysis of residual lignocellulosic biomass, including the analysis of possible applications for tars, which deserved a complete section (Section 3.9).
4) It would be more interesting if the authors focused more on the significance of their findings regarding the importance of the interrelationship between the obtained results and sustainable development/cleaner production in the sector context, and the barriers of doing it, what would be the consequences, in the real world, in changing the observed situation, what would be the ways, in the real world, to change/improve the observed situation.
We discussed the results of our experiments bearing in mind their importance in the determination of possible applications for tars (see Section 3.9). Moreover, the various positive consequences on environmental, social and economic issues expected from the use of pyrolysis processes to valorize residual lignocellulosic biomass were approached in the Introduction section.

Reviewer 3 Report
Comments and Suggestions for Authors
Overall, the manuscript is well-written and provides literature contribution to the field with the analysis on the tars produced. Some minor revision will be needed to polish the manuscript.
1. Line 67: "Effluent streams" term can be changed to bio-products for better understanding.
2. Line 68: Pyrolysis by right should be carried out in an inert condition.
3. The objective of the study should be emphasized at the end of the introduction. The significance of the study should be highlighted.
4. Line 165: What kind of response factors were determined? Please elaborate.
5. Line 181: Please elaborate the Dulong's formula calculation.
Author Response
REVIEWER 3
Overall, the manuscript is well-written and provides literature contribution to the field with the analysis on the tars produced. Some minor revision will be needed to polish the manuscript.
- Line 67: "Effluent streams" term can be changed to bio-products for better understanding.
We appreciate the positive comment by the reviewer. We accepted the reviewer´s suggestion (see line 67).
- Line 68: Pyrolysis by right should be carried out in an inert condition.
One of the most important parameters in biomass thermochemical processes, besides temperature, is the equivalence ratio (ER) defined for gasification. ER is the relationship between the actual amount of air used and the stoichiometric amount for complete combustion. According to numerous studies devoted to biomass thermochemical processes, pyrolysis can be defined as the processes where ER is lower than 0.2 [4, 5]; in our work ER was 0.11, thus fulfilling this definition.
[IV] Li, B.; Song, M.; Xie, X.; Wei, J.; Xu, D.; Ding, K.; Huang, Y.; Zhang, S.; Hu, X.; Zhang, S. & Liu, D. (2023). Oxidative fast pyrolysis of biomass in a quartz tube fluidized bed reactor: Effect of oxygen equivalence ratio. Energy 270, 126987.
[V] Polin, J. P., Carr, H. D., Whitmer, L. E., Smith, R. G., & Brown, R. C. (2019). Conventional and autothermal pyrolysis of corn stover: Overcoming the processing challenges of high-ash agricultural residues. Journal of analytical and applied pyrolysis 143, 104679.
- The objective of the study should be emphasized at the end of the introduction. The significance of the study should be highlighted.
Following the advice by the reviewer, we have now emphasized the objective and the significance of our study in the Introduction section (see lines 123 to125).
- Line 165: What kind of response factors were determined? Please elaborate.
A more detailed description of the procedure to determine response factor was now incorporated in Section 2.3 (see lines 186 to 201)).
- Line 181: Please elaborate the Dulong's formula
A more detailed description of the Dulong's formula was included in the manuscript (see lines 137 to 142).

Round 2
Reviewer 1 Report
Comments and Suggestions for Authors
The authors responded well to the comments and substantially improved the manuscript. It is with great pleasure that I recommend the revised manuscript to be published in the journal Processes.